# OP-LoRA: The Blessing of Dimensionality with Overparameterized Low-Rank Adaptation

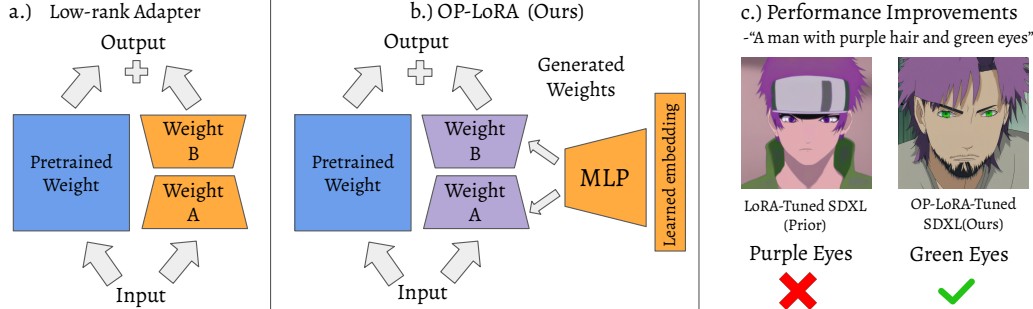

Figure 1: **Comparison of standard LoRA and OP-LoRA**. (a) A standard Low-Rank Adapter learns two rank-reduced matrices (A and B) that are added to the frozen base weights. (b) Our proposed OP-LoRA consists of a reparameterization, where adapter weights are predicted from an MLP and a learned embedding. (c) Qualitative image generation results when adapting Stable Diffusion XL to Naruto (Cervenka, 2022), with OP-LoRA demonstrating higher quality and more faithful reconstruction of the text prompt than standard LoRA.

## ABSTRACT

Low-rank adapters (LoRA) enable finetuning of large models with only a small number of parameters, reducing storage costs and minimizing the risk of catastrophic forgetting. However, they often suffer from an ill-conditioned loss landscape, leading to difficult optimization. Prior work addresses these challenges by aligning adapter updates with full finetuning gradients via custom optimizers, but these methods lack the flexibility to accommodate new adapter architectures and are computationally expensive. We instead introduce OP-LoRA, a novel method which replaces each LoRA adapter with weights predicted by an extra MLP, which is discarded after training. This temporarily allows additional parameters during training to improve optimization, yet requires less wall time than custom optimizers and zero extra cost at inference time because the MLP is discarded. Crucially, extending OP-LoRA to other adapters is as simple as modifying the size of the prediction head for each new adapter type. Since the additional parameters are used only during training and thrown away before inference, there is no risk of overfitting due to increased representational capacity, unlike simply raising the LoRA rank. Instead, we show that this approach allows the optimization to adaptively increase or decrease step size, improving performance and decreasing sensitivity to learning rate. On both small and large-scale LoRA tuning tasks, we observe consistent performance gains of OP-LoRA relative to LoRA and its variants. We achieve especially notable improvements in image generation, with OP-LoRA CMMD scores improving by up to 15 points relative to LoRA.

## 1 INTRODUCTION

Finetuning large foundation models for specific tasks can provide significant performance gains but is computationally intensive, with risks of catastrophic forgetting (Ruiz et al., 2024; Cho et al., 2021a; Biderman et al., 2024). Methods utilizing low-rank adapters (LoRA) (Hu et al., 2022; Hayou

et al., 2024; Zhang et al., 2023; Liu et al., 2024; Nikdan et al., 2024; Meng et al., 2024) address these challenges by modifying the model in rank-constrained ways. This preserves generalization and reduces interference with pretrained weights (Figure 1a). However, low-rank adapters can make optimization harder by creating uneven curvature in the loss landscape (Section 3). Even when optimized with AdamW (Loshchilov & Hutter, 2019; Kingma & Ba, 2015), which preconditions gradients, poorly conditioned loss landscapes can still pose a problem (Das et al., 2024). These issues manifest during LoRA training as high sensitivity to learning rates, as shown by Biderman et al. (2024) and confirmed in Section 3.3.

The most successful attempts to address this issue are custom optimizers such as LoRA-Pro (Wang et al., 2025) and ScaledAdamW (Wang et al., 2024), which aim to align the LoRA update with that of full finetuning. However, in implementation, they are complex and difficult to extend to new LoRA variants (see Limitations of Wang et al. (2025)). For example, adapting these optimizers to DoRA (Liu et al., 2024) is non-trivial due to weight normalization, which complicates the projection of full finetuning gradients. They also tend to be more expensive to run than standard optimizers, requiring matrix inversions and expensive optimizations. In our testing using authors' code, wall time of ScaledAdamW is 15% longer than that of OP-LoRA, and LoRA-Pro is up to 14x longer (Section 4.4) when finetuning LLaMA on Commonsense Reasoning tasks. These limitations highlight the need for alternative optimization strategies that are both effective and architecture-agnostic.

Instead of relying on specialized optimizers, we propose a fundamentally different and more flexible approach, which we call OP-LoRA (Overparameterized Low-Rank Adaptation). OP-LoRA uses a small MLP as a hypernetwork (Ha et al., 2017) to predict the low-rank adapter matrices at **train time only** (Figure 1b). In contrast to other hypernetworks (Ruiz et al., 2024; Ortiz-Barajas et al., 2024), we do not condition on the input sample in any way. This allows us to discard the MLP at inference time, making inference and storage costs equal to that of standard LoRA. OP-LoRA also retains exactly the same representational capacity as standard LoRA: any adapter weights produced by the MLP can be expressed directly with standard LoRA parameters. Despite the higher parameter count during training, the extra parameters do not increase model capacity, decreasing risk of overfitting. We show that this "blessing of dimensionality" is due to an increased ability to navigate complex loss landscapes by an acceleration mechanism (Section 3.2).

Integration of OP-LoRA takes only a few lines of code and generalizes to any LoRA variant. For example, our OP-DoRA extension of DoRA simply adds a second MLP head to predict its extra adapter weights, something which would be difficult to do with custom optimizers. OP-LoRA outperforms standard variants by 1-6% on natural language tasks and up to 15 CMMD points on image generation tasks (Section 4).

We summarize our contributions as follows:

- We introduce OP-LoRA, a novel yet easy-to-implement reparameterization of LoRA that uses an MLP to predict adapter weights instead of learning them directly. After training, the MLP is discarded so zero additional storage or inference costs are incurred.
- We show that OP-LoRA navigates loss landscapes better than standard LoRA due to a built-in acceleration mechanism (Section 3.2).
- We empirically validate OP-LoRA on a large range of tasks including image and text generation and show consistent performance gains on both, and a large improvement in adapting Stable Diffusion relative to standard LoRA (Section 4).

More generally, we believe that train-time over-parameterization represents a promising yet underexplored paradigm in model training, and we hope that our work will catalyze further work.

## 2 RELATED WORK

**Low-rank finetuning:** Low-rank finetuning, specifically with LoRA (Low-Rank Adaptation) (Huh et al., 2023), has emerged as a powerful approach for adapting pre-trained models with minimal additional parameters. A number of follow ups have emerged, improving performance. AdaLoRA (Zhang et al., 2023) prunes weights during training. DoRA (Liu et al., 2024) adds a magnitude scaling vector to the updated matrix. RoSA (Nikdan et al., 2024) adds a sparse weight update to the low-rank update, but requires a full-finetuning pass to compute the weight mask.

OLoRA(Büyükakyüz, 2024) and PiSSA (Meng et al., 2024) ease optimization by initializing LoRA orthogonally. However, they are prone to overfitting due to removing important components from the frozen base weights. Another approach is to make the LoRA optimization trajectory similar to that of full finetuning: LoRA-GA (Wang et al., 2025) initializes LoRA parameters to an SVD approximation of the full-finetuning gradient, while LoRA-Pro (Wang et al., 2024) and ScaledAdamW (Zhang & Pilanci, 2024a) project full-tuning gradients onto the LoRA subspace. Both have computational overhead resulting in extended training times: LoRA-Pro requires expensive computations in gradient projection while LoRA-GA requires a full-finetuning pass similar to RoSA. Deep LoRA (Yaras et al., 2024) learns an over-parameterized LoRA first before compressing it, leveraging the over-parameterization for improved training. However, the compression is an expensive process, and therefore impractical at large scale.

**Reparameterization with hypernetworks:** Ha et al. (2017) generate weights of an LSTM and CNN from a neural network, introducing the concept of HyperNetworks. However, their focus is relaxing weight sharing in LSTMs and reducing parameter count in convolutional networks for image classification. In contrast, we leverage over-parameterization for improved performance. HyperDreamBooth (Ruiz et al., 2024) generates initializations for LoRA parameters from an input image. In contrast, we learn our parameter-generating MLP with a learned parameter vector as input. HyperLoader (Ortiz-Barajas et al., 2024) uses hypernetwork to generate adapters, but share parameters between layers and tasks. We show that this shared structure severely reduces performance in Table 3.

**Convergence Properties of Neural Networks:** Understanding the convergence behavior of neural networks has been a subject of significant research interest(Du et al., 2019; Nguyen & Hein, 2017). Du et al. (2019) find that gradient descent can find global optima in ResNets. Huang et al. (2020) find that optimizers are biased to flat minima in overparameterized models, and coin the term the "blessing of dimensionality". Most relevant to OP-LoRA, though, is Arora et al. (2018)'s work showing that stacking linear layers can function as an implicit acceleration mechanism in gradient descent.

# 3 METHODOLOGY: OVERPARAMETERIZED LORA (OP-LORA)

Low-Rank Adaptation (LoRA) has become a popular strategy for finetuning large models, allowing adaptation to new tasks by learning a low-rank matrix factorization of weight updates. With LoRA, finetuning a model layer's weight matrix $W_0 \in \mathbb{R}^{d \times d}$ is achieved by learning an additive low-rank update $\Delta W$ such that the adapted weights $W$ are given by:

$$W = W_0 + \Delta W = W_0 + BA, \tag{1}$$

where $A \in \mathbb{R}^{r \times d}$ and $B \in \mathbb{R}^{d \times r}$ are learned low-rank matrices, with $r \ll d$, reducing the number of parameters to learn. However, LoRA introduces challenges during optimization. While the original parameter space has curvature defined by the Hessian $H_W$, the LoRA $A$ matrix has the transformed Hessian, show as a composition of functional operators:

$$H_A = B^\top \circ H_W \circ B. \tag{2}$$

This transformation affects the condition number of the optimization problem in the $A$-space. Even if the original Hessian $H_W$ is well-conditioned and symmetric positive definite (a reasonable assumption near local minima), the reparameterized Hessian can become ill-conditioned depending on the singular values of $B$. In particular, the condition number $\kappa(H_A)$ satisfies:

$$\max \left\{ \frac{\kappa(H_W)}{\kappa(B)^2}, \frac{\kappa(B)^2}{\kappa(H_W)} \right\} \ \leq \ \kappa(H_A) \ \leq \ \kappa(B)^2 \cdot \kappa(H_W) \tag{3}$$

where $\kappa(\cdot)$ denotes the spectral condition number. A higher condition number indicates a greater ratio between the largest and smallest curvatures of the loss landscape. In practice, higher condition numbers lead to slower convergence and greater sensitivity to learning rates. These bounds imply that even if $H_W$ is well conditioned, poor conditioning in $B$ alone can lead to difficulty in optimizing $A$, since they drive up both upper and lower bounds. Biderman et al. (2024) observe that LoRA is sensitive to learning rate, which we confirm in Section 3.3. The full derivation of this result, and a symmetric one for optimizing $A$, is provided in the Appendix. In Section 3.2 and 3.3, we show OP-LoRA can dynamically adjust step size and do an adaptive line search, overcoming the optimization difficulties of standard LoRA.

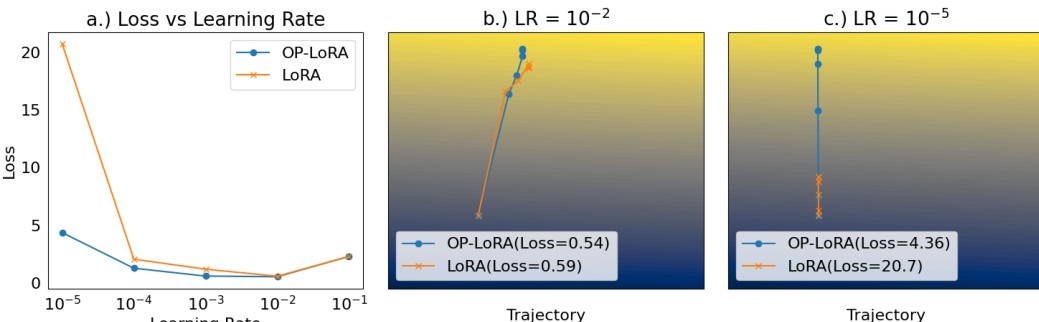

Figure 2: **Optimization behavior of LoRA and OP-LoRA for Rotated MNIST classification**: **a.)** Training loss achieved by LoRA and OP-LoRA as a function of the learning rate, showing that OP-LoRA attains lower loss across a wide range of learning rates and remains robust even at suboptimal step sizes. **(b)** and **(c)** Parameter-update trajectories overlaid on the training loss surface at (b) the optimal learning rate and (c) a low learning rate. In both cases, OP-LoRA descends more directly towards lower loss (yellow contours) than LoRA.

### 3.1 PREDICTING LoRA WEIGHTS

In order to avoid the optimization difficulties of training LoRA discussed in Section 3, we avoid directly optimizing $A$ and $B$ by introducing a two-layer MLP which takes as input $z$ and predicts the entries of $A$ and $B$. This is a way of reshaping the optimization landscape, making it easier to optimize. However, because the additional parameters are discarded after training, they **do not increase finetuning capacity**. This is an important distinction with other methods of adding parameters to the finetuning procedure like increasing LoRA rank. Concretely, we generate $A$ and $B$ as flattened matrices via:

$$\begin{pmatrix} A \\ B \end{pmatrix} = \mathbf{W}_2(\text{ReLU}(\mathbf{W}_1 z + c_1)) + c_2 \tag{4}$$

where $z$ is the learned input vector to the MLP, $\mathbf{W}$ and $c$ correspond to learned weights biases, and $A \in \mathbb{R}^{r \times d}$ and $B \in \mathbb{R}^{d \times r}$ are the generated matrices.

Once finetuning is complete, the MLP can be discarded, retaining only the low-rank matrices $A$ and $B$ for inference and storage. Furthermore, as in standard LoRA, $A$ and $B$ can be merged with the pre-trained model's weights by adding $\Delta W = BA$ to the relevant layer weights (Equation 1). Although the MLP is compact in depth, it predicts a high-dimensional output, the size of the LoRA parameters, increasing its parameter count. For instance, an MLP with a hidden dimension of 32 scales the number of trainable parameters by approximately 32. This makes OP-LoRA particularly advantageous in settings where inference resources are constrained, but sufficient memory is available during training. Further, because the MLP is small relative to a typical base model, wall-time penalties are not large. For more details on computational cost, see Section 4.4.

### 3.2 OPTIMIZATION BENEFITS OF OP-LoRA

Arora et al. (2018) prove that increasing depth by replacing linear layers with products of matrices, which has the same expressive power as a single matrix due to the linear nature of the transformation, leads to faster convergence. Although we focus on re-parameterizing with an MLP instead of increasing depth, we employ the same theoretical framework to examine OP-LoRA's enhanced training dynamics below. While we consider the linear case here for clarity, we extend the analysis to the MLP case in the Appendix.

Consider the OP-LoRA reformulation of parameter vector $v$ with a two layer MLP, defined as $v = \mathbf{W}_2(\text{ReLU}(\mathbf{W}_1 z + c_1)) + c_2$. We can then assign vector $h = \text{ReLU}(\mathbf{W}_1 z + c_1)$, and for clarity of derivation, we treat $h$ as a free parameter vector; this corresponds to only updating the bias in the first layer of the MLP. We also merge bias parameters into the parameter matrices $\mathbf{W}$ for ease of notation, assuming a constant value is appended to $z$ and $h$. This leaves us with simpler

reparameterization form $v = \mathbf{W}_2 h$, where $v \in \mathbb{R}^p, \mathbf{W}_2 \in \mathbb{R}^{p \times k}$, $h \in \mathbb{R}^k$ and $p$ is the size of generated parameter vector $v$ and $k$ is the hidden dimension of the MLP re-parameterization.

The update rule for $\mathbf{W}_2$ and $h$ with learning rate $\eta$ is given by:

$$\mathbf{W}_2^{(t+1)} = \mathbf{W}_2^{(t)} - \eta \nabla_{\mathbf{W}_2}, \quad h^{(t+1)} = h^{(t)} - \eta \nabla_h. \tag{5}$$

Then, the parameter vector $v$ at time step $t + 1$ becomes:

$$v^{(t+1)} = \mathbf{W}_2^{(t+1)} h^{(t+1)} = \left( \mathbf{W}_2^{(t)} - \eta \nabla_{\mathbf{W}_2} \right) \left( h^{(t)} - \eta \nabla_h \right). \tag{6}$$

Expanding the product, we have:

$$v^{(t+1)} = \mathbf{W}_2^{(t)} \mathbf{h}^{(t)} - \eta \nabla_{\mathbf{W}_2} h^{(t)} - \eta \mathbf{W}_2^{(t)} \nabla_h + \eta^2 \nabla_{\mathbf{W}_2} \nabla_h. \tag{7}$$

Following (Arora et al., 2018; Zhang & Pilanci, 2024b), we ignore the higher order term since the learning rate $\eta$ is assumed to be small and therefore the term shrinks to 0. By definition, $v = \mathbf{W}_2 h$, so:

$$v^{(t+1)} = v^{(t)} - \eta \nabla_{\mathbf{W}_2} h^{(t)} - \eta \mathbf{W}_2^{(t)} \nabla_h \tag{8}$$

By substituting chain-rule expansions of $\nabla_{\mathbf{W}_2}$ and $\nabla_h$, the new value for $v$ becomes:

$$\nabla_{\mathbf{W}_2} = \nabla_v h^T, \quad \nabla_h = \mathbf{W}_2^T \nabla_v. \tag{9}$$

$$v^{(t+1)} = v^{(t)} - \underbrace{\eta \, \|h^{(t)}\|^2 \nabla_{v^{(t)}}}_{\text{trainable learning rate}} - \underbrace{\eta \, \mathbf{W}_2^{(t)} \big( (\mathbf{W}_2^{(t)})^T \nabla_v \big)}_{\text{adaptive line search}} \tag{10}$$

This reveals two key properties of the optimization trajectory under OP-LoRA. First, OP-LoRA introduces a dynamic learning rate scaling factor, $\|h^{(t)}\|^2$. From the update rule $\nabla_h = (\mathbf{W}_2^{(t)})^T \nabla_v$, we see that $\|h^{(t)}\|^2$ grows when the gradient is positively aligned with $\mathbf{W}_2^{(t)}$ and shrinks otherwise. In the special case where $k = 1$, the vector $h$ becomes a scaling factor for the single column vector $\mathbf{W}_2$ which represents the direction of parameter vector $v$. Consequently, if the optimizer overshoots a minimum, the sign of the gradient update for $h$ flips, and the effective learning rate decreases, but if consecutive updates align, it increases. In the more general $k > 1$ setting, there are $k$ such scalar factors (one for each column in $\mathbf{W}_2^{(t)}$) and each can increase or decrease independently, and the overall learning rate becomes scaled by $\|h^{(t)}\|^2$ . We refer to $\|h^{(t)}\|^2$ as the ***trainable learning rate***.

Second, OP-LoRA adds an extra update term $\mathbf{W}_2^{(t)} \big( (\mathbf{W}_2^{(t)})^T \nabla_v \big)$, which shifts parameters along the already learned directions $\mathbf{W}_2^{(t)}$, in proportion to the gradient's projection onto those directions. When $k = 1$, this can be seen as a gradient step in a line-search, along the direction of the current state of $v$ , and naturally biases the updates toward directions already taken. However, if $\nabla_v$ suddenly changes to a new and orthogonal direction, the final term immediately vanishes, causing the effective step size to shrink right away and the update to suddenly shift towards the current gradient. For $k > 1$, $v$ becomes a weighted sum over each column of $\mathbf{W}_2^{(t)}$, and the parameter update is biased toward any directions in $\mathbf{W}_2^{(t)}$ on which the gradient has nonzero projection. We refer to $\mathbf{W}_2^{(t)} \big( (\mathbf{W}_2^{(t)})^T \nabla_v \big)$ as ***adaptive line search***. Together, the ***adaptive line search*** and ***trainable learning rate*** suggest that OP-LoRA has an improved ability to navigate complex loss landscapes. While the overall trainable learning rate dynamically updates to the given problem, the adaptive line search can rapidly search the subspace spanned by $W_2$. In Section 3.3, we explore how this improves performance in a small-scale setting.

## 3.3 MNIST Case Study

To illustrate the advantages of OP-LoRA, we start with a small scale study on MNIST, showing that OP-LoRA converges to better loss and also is less sensitive to learning rate.

We use a two-layer MLP $f(x)$ with hidden dimension 512. We train $f$ on MNIST for 30 epochs to nearly 0 training loss. This constitutes our base model, which we freeze before continued LoRA tuning on Rotated MNIST to create an adaptation task. We finetune the frozen $f$ with LoRA and OP-LoRA adapters of rank $r = 4$, but this time on the new task of Rotated MNIST, where each MNIST sample is randomly rotated before classification.

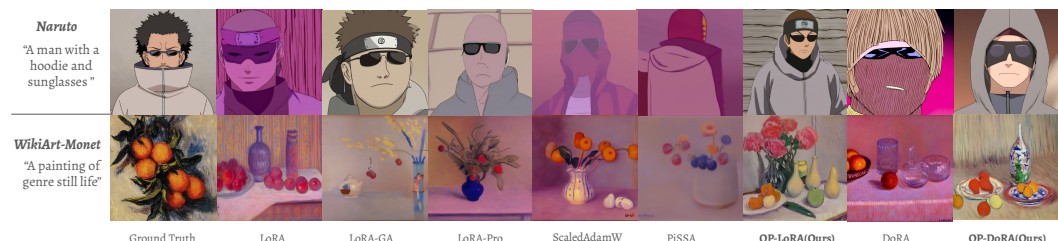

Figure 3: Comparison of generated images across different low-rank finetuning methods of SD-XL for two datasets: Naruto (upper) and Claude Monet-style painting (lower). Each row shows outputs from a specific model configuration based on ground truth captions. For the Naruto prompt, OP-LoRA and OP-DoRA effectively capture the presence of a hoodie, and generally generate higher fidelity images. For the Monet-style paintings, OP-LoRA and OP-Dora offer more realistic scenes.

We find two main results. First, we find that OP-LoRA achieves lower training loss than LoRA. In Figure 2, we can see that train loss for LoRA reaches 0.59, vs 0.54 for OP-LoRA. Second, we find that OP-LoRA is much less sensitive to learning rates than LoRA (Figure 2 a.), with losses staying relatively low even with learning rates two to three orders of magnitude smaller than optimal. The learning rate sensitivity of standard LoRA corroborates to findings in (Biderman et al., 2024) (See Figure of Biderman et al. (2024)). Qualitatively, we plot the optimization trajectory of both LoRA and OP-LoRA in Figure 2 b.) and c.). Interestingly, for learning rates which are three orders of magnitude lower than optimal, OP-LoRA makes good progress, demonstrating OP-LoRA's ability to accelerate training. We highlight that this is despite both methods being optimized with Adam, an optimizer with adaptive learning rates and a momentum term to help with slow learning.

We do additional analysis on the gradient properties of the Hessian using power iteration in Appendix B.1, where we find that gradient norm in the direction of highest curvature is much higher for OP-LoRA than standard LoRA.This empirically suggests that OP-LoRA may be less sensitive to poor conditioning than LoRA, since it can minimize loss effectively even when high curvature requires a small learning rate. This is consistent with the view that OP-LoRA adaptively reshapes the LoRA loss landscape for better and faster optimization through reparameterization.

## 4 EXPERIMENTS

In this section, we show performance on three different tasks. We start by finetuning Stable Diffusion (Podell et al., 2024) in Section 4.1. Low-rank finetuning offers particular advantages in this context, especially since individual users often have only a few images for model specialization, making regularization with low-rank updates particularly valuable. Additionally, storage concerns are significant, as users may want to save numerous specialized model variants. This means that it is beneficial for any over-parameterization scheme to be training only, to minimize storage costs for the end user. In Section 4.2, we move onto visual question answering with a VL-Bart (Cho et al., 2021a) model, where we demonstrate that OP-LoRA and OP-DoRA show consistent improvements over their standard counterparts. Finally, we show results of a finetuned LLaMA (Touvron et al., 2023) model on CommonSense reasoning tasks in Section 4.3. This is another common PEFT use-case, where full finetuning is too expensive and fully-fine tuned models become cumbersome to store. Details beyond what is provided below are in the Appendix. Finally, in Appendix B , we show results for improving VeRA(Kopiczko et al., 2024),Mix-of-Show(Gu et al., 2023), and Matrix Factorization with MLP over-parameterization.

### 4.1 FINETUNING STABLE DIFFUSION

We finetune Stable Diffusion XL (Podell et al., 2024) for two datasets, a Claude Monet subset of Wiki-Art (Face & Huggan, 2023) and Blip-Captioned Naruto Images (Cervenka, 2022), and evaluate using MMD distances over CLIP embeddings.

**Datasets: WikiArt (Face & Huggan, 2023)** is a dataset of around 80000 pieces of artwork; each labeled with artist, genre, and style. We filter by the artist Claude Monet, leaving 1334 images, and

| Method | Naruto | WArt | Avg ($\downarrow$) |
|---|---|---|---|
| *W/ grad. alignment* | | | |
| LoRA-GA$_{r=4}$ (Wang et al., 2024) | **15.2** | 43.7 | 29.5 |
| LoRA-Pro$_{r=4}$ (Wang et al., 2025) | 20.9 | 32.4 | 26.7 |
| SAdamW$_{r=4}$ (Zhang & Pilanci, 2024b) | 21.9 | **30.5** | **26.2** |
| *W/out grad. alignment* | | | |
| PiSSA$_{r=4}$ (Meng et al., 2024) | 29.7 | 43.2 | 36.5 |
| LoRA$_{r=4}$ (Hu et al., 2022) | 23.7 | 47.9 | 35.8 |
| DoRA$_{r=4}$ (Liu et al., 2024) | 17.2 | 49.8 | 33.5 |
| OP-DoRA$_{r=4}$ (ours) | 11.9 | 46.6 | 29.3 |
| OP-LoRA$_{r=4}$ (ours) | **9.6** | **31.7** | **20.7** |

Table 1: Finetuning Stable Diffusion: CMMD on Naruto and WikiArt (lower is better).

| Method | VQAv2 | GQA | NLVR | Avg ($\uparrow$) |
|---|---|---|---|---|
| Full FT | 66.9 | 56.7 | 73.7 | 65.8 |
| LoRA$_{r=128}$ (Hu et al., 2022) | 65.5 | 53.9 | 72.0 | 63.8 |
| OP-LoRA$_{r=128}$ (ours) | **65.6** | **54.9** | **73.0** | **64.5** |
| DoRA$_{r=128}$ (Liu et al., 2024) | 65.8 | 54.9 | 72.4 | 64.4 |
| OP-DoRA$_{r=128}$ (ours) | **66.4** | **55.1** | **74.0** | **65.2** |

Table 2: VQA evaluation with VL-BART, measuring accuracy. OP-LoRA and OP-DoRA outperform their non-overparameterized counterparts by around 1%.

construct text captions for finetune of the form 'A painting of genre $\langle \rangle$', for example 'A painting of genre *portrait*'. We found a single low-rank adapter to not be able to model many artists jointly. Importantly, the artist name Claude Monet is not mentioned in the caption, so the Claude Monet style has to be learned from the data. **Naruto BLIP Captions (Cervenka, 2022)** is a dataset of 1221 anime images from the Japanese manga series Naruto, which are captioned by BLIP (Li et al., 2022). **Finetuning protocol:** We train for two epochs and we target only the attention layers in the U-Net in Stable Diffusion XL 1.0. We use rank $r = 4$ and OP-LoRA MLP width 32. We choose a number of baselines in addition to LoRA intended to make optimization easier. LoRA-GA (Wang et al., 2024), LoRA-Pro (Wang et al., 2025) and (Zhang & Pilanci, 2024b) ScaledAdamW leverage full finetuning gradient information, while PiSSA (Meng et al., 2024) initializes to principle components. DoRA adds a weight normalization to LoRA in order to improve the learning ability of LoRA. We extend OP-LoRA to OP-DoRA by adding an additional prediction head to generate the weight scaling factors along side the low-rank matrices.

**Evaluation Protocol:** For evaluation, we aim to measure how different the distribution of generated images is from the ground truth distribution. We generate a new image for each training caption. To assess the quality of the generation, we compute the CLIP Maximum Mean Discrepancy (Jayasumana et al., 2024) (CMMD) distance, which computes MMD over clip scores. Lower CMMD values indicate lower distributional distances between generated samples and the ground truth, and therefore higher quality generations. Jayasumana et al. (2024) show CMMD to be a better measure of generated image quality than alternatives such as FID Score or Inception Score, aligning better with human raters and providing more consistent results with varied sample sizes.

**Results:** We present CMMD scores in Table 1. Two interesting trends emerge. First, OP-LoRA outperforms OP-DoRA on both the Naruto and WikiArt datasets. We attribute this to the increased ease of overfitting with DoRA. Second, OP-LoRA and OP-DoRA, achieve substantially improved scores over their standard counterparts. Specifically, OP-LoRA achieves a CMMD score of 9.6 on the Naruto dataset, compared to 23.7 for LoRA, and similarly, OP-DoRA scores 11.9 compared to DoRA's 17.2. On the WikiArt dataset, OP-LoRA also shows a substantial gain with a score of 31.7 compared to 47.9 for LoRA. Furthermore, OP-LoRA outperforms other baseline methods on average, including state-of-the-art optimizers such as LoRA-Pro (Wang et al., 2025) and ScaledAdamW (Zhang & Pilanci, 2024b) which leverage information about the full-finetuning gradient. This suggests that following the full-finetuning gradient as closely as possible is not the only way for parameter efficient adapters to perform well, and different approaches are worth exploring.

We also show samples of generated images in Figure 3, where we compare LoRA to OP-LoRA and DoRA to OP-DoRA. We can overall see much higher quality for the over-parameterized variants. For example, OP-LoRA and OP-DoRA capture the hoodie, while LoRA and DoRA do not. The still life setting for OP-LoRA is more complex, with flowers. Finally, DoRA seems to generate a somewhat degenerate image in the second row, while OP-LoRA and OP-DoRA do not. We provide many random samples in the Appendix.

## 4.2 VISUAL QUESTION ANSWERING EXPERIMENTS

**Datasets: VQAv2** (Goyal et al., 2017) (113K training images) and **GQA** (Hudson & Manning, 2019) (82.7K training images) are both visual question answering datasets scored. **NLVRv2** (Suhr et al.,

| Method | Train Par. | Inf Par. | BoolQ | PIQA | SIQA | HSwag | WinoG | ARC-E | ARC-C | OBQA | AVG |
|---|---|---|---|---|---|---|---|---|---|---|---|
| *W/ grad alignment* | | | | | | | | | | | |
| LoRA-GA$_{r=32}$ (Wang et al., 2024) | 0.83 | 0.83 | 63.0 | 73.4 | 75.5 | 53.0 | 74.9 | 66.4 | 51.7 | 70.8 | 66.1 |
| LoRA-Pro$_{r=32}$ (Wang et al., 2025) | 0.83 | 0.83 | 69.6 | 81.6 | 77.7 | **84.4** | **80.3** | 81.7 | 65.5 | **80.2** | 77.6 |
| SAdamW$_{r=32}$ (Zhang & Pilanci, 2024b) | 0.83 | 0.83 | **70.7** | **82.3** | **78.2** | 83.3 | 79.6 | **82.6** | **66.4** | 78.6 | **77.7** |
| SAdamW$_{r=16}$ (Zhang & Pilanci, 2024b) | 0.41 | 0.41 | 69.9 | 81.5 | 77.3 | 82.5 | 79.2 | **81.4** | 65.2 | 76.8 | 76.7 |
| OP-SAdamW$_{r=16}$(Ours) | 13.5 | 0.41 | **69.9** | 81.9 | 77.8 | 84.6 | 81.1 | 80.1 | **65.5** | 80.4 | 77.7 |
| *W/Out grad alignment* | | | | | | | | | | | |
| DeepLoRA$_{r=32}$ (Yaras et al., 2024) | 0.83 | 0.83 | 66.7 | 78.8 | 59.8 | 39.6 | 51.7 | 41.6 | 32.8 | 39.8 | 51.4 |
| HLoader$_{r=32}$ (Ortiz-Barajas et al., 2024) | 0.83 | 0.83 | 61.5 | 48.5 | 43.5 | 36.3 | 54.3 | 26.2 | 32.1 | 31.8 | 41.8 |
| AdaLoRA$_{r=32}$ (Zhang et al., 2023) | 1.25 | 0.83 | 67.4 | 80.7 | 77.0 | 47.3 | 79.6 | 81.4 | 64.8 | 76.2 | 71.8 |
| DoRA$_{r=32}$ (Liu et al., 2024) | 0.84 | 0.84 | 65.3 | 65.6 | 76.9 | 81.2 | 78.8 | 79.4 | 64.0 | 78.3 | 73.7 |
| LoRA$_{r=32}$ (Hu et al., 2022) | 0.83 | 0.83 | 67.5 | 80.8 | **78.2** | 83.4 | 80.4 | 78.0 | 62.6 | 79.1 | 76.3 |
| OP-LoRA(Ours)$_{r=32}$ | 27.4 | 0.83 | **69.0** | 81.4 | 77.9 | 85.7 | 79.2 | 80.5 | 64.4 | 78.6 | 77.1 |
| OP-DoRA(Ours)$_{r=32}$ | 27.7 | 0.84 | 67.2 | **82.0** | 76.3 | **86.5** | 81.4 | 81.5 | 65.3 | 80.3 | 77.5 |
| LoRA$_{r=16}$ (Hu et al., 2022) | 0.41 | 0.41 | 69.9 | 77.8 | 75.1 | 72.1 | 55.8 | 77.1 | 62.2 | 78.0 | 70.9 |
| LoRA$_{r=466}$ (Hu et al., 2022) | 12.1 | 12.1 | 53.6 | 70.1 | 73.4 | 52.3 | 71.3 | 61.4 | 46.0 | 64.6 | 61.6 |
| OP-LoRA(Ours)$_{r=16}$ | 13.5 | 0.41 | **70.3** | **83.1** | 77.7 | 77.7 | 79.1 | 78.9 | 63.9 | 78.8 | 76.2 |

Table 3: Finetuning of LLaMA 7B (Touvron et al., 2023) on commonsense reasoning datasets. OP-LoRA and OP-DoRA outperform their standard counterparts, and are on par with the complex custom optimizers such as LoRA-Pro(Wang et al., 2025) and ScaledAdamW (Zhang & Pilanci, 2024b) despite not leveraging information about the full FT gradient. When combined with ScaledAdamW, OP-LoRA can match standard ScaledAdamW performance with half the inference parameters.

2019) (206K training images) is a visual reasoning dataset, answering a True/False question about a pair of images. All are evaluated with accuracy.

**Finetuning and evaluation protocol:** We follow Liu et al. (2024) in our finetuning protocol. We fine-tune VL-BART (Cho et al., 2021b) in multi-task way. VL-BART composes a CLIP-ResNET101 (Radford et al., 2021) and Bart $_{Base}$ (Lewis et al., 2020) model, and trains all tasks jointly with a language-modeling loss. We finetune for 20 epochs with rank $r = 128$, targeting only $Q$ and $K$ matrices in attention layers, while also training biases. We use MLP hidden layers size 4, based on our experiments in Section 4.4.

**Results:** We present results in Table 2. We can reach a similar conclusion for the vision-language task as for the image generation task; OP-LoRA and OP-DoRA improve over their counterparts. The roughly 1% improvement achieved by the OP variants matches the gains from LoRA to DoRA.

## 4.3 COMMONSENSE REASONING EXPERIMENTS

**Datasets:** The Commonsense task consists of 8 sub-tasks, with about 170k training sequences total. **BoolQ** (Clark et al., 2019) is a yes/no question-answering dataset. **PIQA** (Bisk et al., 2020) requires physical knowledge to answer. **SIQA** (Sap et al., 2019) is about social reasoning for humans. **HellaSwag** (Zellers et al., 2019) asks the model to complete the context with a sentence. **WinoGrande** (Sakaguchi et al., 2021) asks the model to fill in the blank. **ARC-E** and **ARC-C** (Clark et al., 2018) are easy and hard variants of multiple choice science questions. **OBQA** (Mihaylov et al., 2018) asks multiple choice questions requiring strong comprehension skills of context.

**Finetuning and evaluation protocol:** We follow Liu et al. (2024) and train with all datasets jointly for 3 epochs, but evaluate each dataset independently. We use $r = 32$ and MLP width 32. In addition to baselines intended to make optimization easier to optimize by leveraging full finetuning gradient information (LoRA-GA (Wang et al., 2024), LoRA-Pro (Wang et al., 2025), SAdamW (Zhang & Pilanci, 2024b) and OP-SAdamW), we also compare to the AdaLora (Zhang et al., 2023), DeepLoRA (Yaras et al., 2024), and HyperLoader (Ortiz-Barajas et al., 2024). OP-SAdamW adds MLP overparameterization to OP-LoRA by inserting gradient projections from ScaledAdamW to the backward pass into the generating MLP. AdaLoRA dynamically allocates parameters to different layers, useful for PEFT training of large scale models. Like OP-LoRA, DeepLoRA (Yaras et al., 2024) is motivated by overparameterization. In its conception, it trains an over-parameterized adapter and compresses it, but this process is too expensive for large models. Therefore, Yaras et al. (2024) simply add a third square matrix to the LoRA adapter, therefore training a product of three matrices. HyperLoader is similar to OP-LoRA, but shares parameters

| Method | GPU Mem | Time |
|---|---|---|
| LoRA (Hu et al., 2022) | 44 GB | 3.5 H |
| OP-LoRA | 69 GB | 4 H |
| ScaledAdamW (Zhang & Pilanci, 2024b) | 44 GB | 4.5 H |
| LoRA-Pro (Wang et al., 2025) | 46 GB | 56 H |

Table 4: GPU Memory and wall time cost, evaluated on CommonSense Reasoning on an H100 HBM3 GPU. The increased memory usage is manageable, and Wall Time is faster than alternatives.

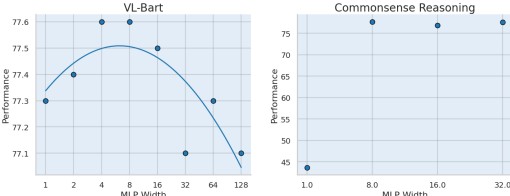

Figure 4: Effect of MLP hidden layer size for OP-DoRA. Performance follows an inverted U-shape for VL-Bart. For Commonsense reasoning, size 1 is too little but otherwise the trend is flat.

between LoRA-generating MLPs, and is used to test the necessity of decoupling parameters. Finally, we expand the rank of LoRA to 466, to verify that performance gains are not simply from naive adding of parameters.

**Results** Table 3 presents the results of our experiments, where both OP-LoRA and OP-DoRA consistently outperform their non-overparameterized counterparts by a margin of 1-4%. Moreover, OP-LoRA nearly matches recent LoRA variants such which align with full finetuning such as LoRA-Pro and ScaledAdamW, but with lower training time. LoRA-Pro takes 56 hours to complete compared to the 4 hours for OP-LoRA. Furthermore, we emphasize that although ScaledAdamW and LoRA-Pro slightly outperform OP-LoRA on commonsense reasoning tasks, they substantially under perform on image generation tasks (Section 4.1) and that combining ScaledAdamW with OP-LoRA enables us to reach the same performance but with **half the inference parameters due to lower rank**. This enables substantially lower inference storage costs. It also decreases inference costs in scenarios where many LoRAs are served concurrently(Sheng et al., 2023). Interestingly, we can see that HyperLoader, which is like OP-LoRA but shares MLP parameters between layers, performs very poorly in our single task context. This supports our design choice to decouple parameters between LoRA adapters, and that the subspace spanned by $W_2$ of the OP-LoRA MLP cannot be the same between adapters. Finally, we can see that expanding rank to 466 is not as performant as a parameter equivalent OP-LoRA, verifying that HOW we add parameters is important, and not JUST that we add parameters.

### 4.4 OP-LoRA Analysis

**Computational Costs:** OP-LoRA introduces extra train-time parameters that are thrown away at inference, so there's no added deployment cost. We evaluate computational cost finetuning LLaMA-7B for CommonSense reasoning, for 3 epochs. On an H100 HBM3 GPU with adapter rank = 32, standard LoRA uses 44 GB of GPU memory, whereas OP-LoRA uses 69 GB. The MLP reparameterization slows training by only about 15%, raising wall-clock time for the entire training run from 3.5 h to 4 h on the CommonSense benchmark. By contrast, ScaledAdamW takes around 4.5 h, and LoRA-Pro's heavier computations extend training to around 56 h. Given that OP-LoRA achieves consistently higher performance than LoRA, we believe this to be manageable. Meanwhile, it is around 10% faster than ScaledAdamW and more than 10 times faster than LoRA-Pro.

**Ablating MLP Width:** One natural question is how much to over-parameterize low rank adapters. We study this question for OP-DoRA on the VL-Bart vision-language and commonsense reasoning tasks, by varying MLP hidden layer size. In Figure 4 we see an inverted U-shape for VL-BART; too little over-parameterization is not enough but too much starts degrading performance. For commonsense benchmarks, we see a low score for hidden layer size 1 but otherwise little variation.

### 5 Conclusion

In this work, we introduce OP-LoRA, an MLP-based reparameterization of LoRA. By leveraging over-parameterization we accelerate training without additional inference overhead. Our experiments across diverse tasks demonstrate that OP-LoRA consistently improves performance over LoRA. We believe that train-time over-parameterization represents a promising yet underexplored paradigm in model training, and we hope that our work will inspire broader investigation into its applications.

**Reproducibility Statement:**

We provide code in the supplementary implementing OP-LoRA and additional training details are presented in the Appendix C.

**Ethics Statement:**

OP-LoRA aims to improve the predictive performance of LoRA. This is a general goal that we believe does not require any special ethical consideration. Nevertheless, machine learning models are a tool which can be used for good or ill, and we encourage users of OP-LoRA to consider the implications of any systems built.

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

## A  THEORETICAL ANALYSIS

### A.1  EXTENSION OF OVERPARMETERIZATION ANALYSIS TO MLP

Although our analysis of the optimization benefits of OP-LoRA is carried out in the linear case (Section 3.2 of the main paper), the same principles extend to deeper ReLU networks. We formalize this below.

**Lemma A.1** (First-order expansion of ReLU activations). *Let $h = \mathrm{ReLU}(a)$ with $a = W_1 z$. For a perturbation $\Delta a$, define the activation mask $\mathbf{M} = \mathbf{1}_{\{W_1 z > 0\}}$. Then a first-order Taylor expansion gives*

$$h_{new} \approx h + \mathbf{M} \odot \Delta a. \tag{11}$$

**Lemma A.2** (Expansion of pre-activation perturbations). *Let $a = W_1 z$. For small perturbations $\Delta W_1, \Delta z$,*

$$a + \Delta a = (W_1 + \Delta W_1)(z + \Delta z). \tag{12}$$

*Expanding and subtracting $a$ yields*

$$\Delta a = \Delta W_1 z + W_1 \Delta z + \Delta W_1 \Delta z. \tag{13}$$

**Lemma A.3** (Form of gradient updates). *With gradients $\nabla_h, \nabla_z$, the updates take the form*

$$\Delta W_1 = -\eta (\nabla_h \odot \mathbf{M}) z^\top, \qquad \Delta z = -\eta \nabla_z. \tag{14}$$

*Substituting into the expansion of $\Delta a$ and dropping second order terms gives*

$$\Delta a \approx -\eta ((\nabla_h \odot \mathbf{M}) z^\top z + W_1 \nabla_z). \tag{15}$$

**Theorem A.4** (Approximation of activation perturbations). *Combining the previous results,*

$$\Delta h \approx -\eta \mathbf{M} \odot \left[ (\nabla_h \odot \mathbf{M}) z^\top z + W_1 \nabla_z \right]. \tag{16}$$

**Theorem A.5** (Update rule in the ReLU case). *Let $v$ be the parameter vector defined as in the main paper. Then the update rule is*

$$v^{(t+1)} = v^{(t)} - \eta \|h^{(t)}\|^2 \nabla_{v^{(t)}} - \eta W_2^{(t)} \left( \mathbf{M} \odot \left[ (W_2^\top \nabla_v \odot \mathbf{M}) z^\top z + W_1 \nabla_z \right] \right). \tag{17}$$

**Corollary A.6** (Comparison to the linear case). *In the linear case (Section 3.2 of the main paper), the update reduces to*

$$v^{(t+1)} = v^{(t)} - \eta \|h^{(t)}\|^2 \nabla_{v^{(t)}} - \eta W_2^{(t)} (W_2^{(t)\top} \nabla_v). \tag{18}$$

*Thus both cases share two key terms:*

- ***Trainable learning rate:*** *$-\|h^{(t)}\|^2 \nabla_v$, unchanged from the linear case.*

- ***Adaptive line search:*** *an update along the subspace spanned by the current columns of $W_2$.*

*Remark A.7* (Geometric interpretation in the ReLU case). The adaptive line search retains the same geometric role as in the linear case: shifting updates along the span of $W_2$. However, the ReLU non-linearity induces a *composite over-parameterization*: the values $h$ themselves are generated through an extra layer with nonlinearity, and each column of $W_2$ only contributes when its corresponding ReLU unit is active. This leads to more diverse update directions when different units activate across inputs or training steps.

### A.2 Curvature Analysis

#### A.2.1 LoRA Hessian

We begin with the LoRA reparameterization

$$W = W_0 + BA, \qquad B \in \mathbb{R}^{d \times r}, \ A \in \mathbb{R}^{r \times d}, \ W_0 \in \mathbb{R}^{d \times d}. \tag{19}$$

Let $L(W)$ denote the loss, and let $H_W$ be the Hessian of $L$ at $W_0$, viewed as a linear operator mapping perturbations in $W$ to second–order variations of the loss.

**Lemma A.8** (Quadratic approximation). *For any small perturbation $\Delta W$, a second–order Taylor expansion gives*

$$L(W_0 + \Delta W) \approx L(W_0) + \langle \nabla_W L, \Delta W \rangle + \tfrac{1}{2} \langle \Delta W, H_W(\Delta W) \rangle. \tag{20}$$

**Theorem A.9** (Effective Hessian with respect to $B$). *Fix $A$ and consider variations in $B$. For a perturbation $\Delta B$, we have $\Delta W = \Delta B A$. The corresponding effective curvature operator is*

$$H_B(\Delta B) = H_W(\Delta B A) A^\top. \tag{21}$$

*Proof.* Substitute $\Delta W = \Delta BA$ into the quadratic form:

$$\frac{1}{2}\langle \Delta W,\ H_W \Delta W \rangle = \frac{1}{2}\langle \Delta BA,\ H_W(\Delta BA)\rangle$$
$$= \frac{1}{2}\langle \Delta B,\ H_W(\Delta BA)\,A^\top \rangle,$$

which establishes the claim. $\square$

**Corollary A.10** (Operator and matrix forms of $H_B$). *In operator notation,*

$$H_B = (\cdot\, A^\top) \circ H_W \circ (\cdot A),$$

*where $(\cdot X)$ denotes right multiplication by $X$. In matrix form,*

$$H_B = (A^\top \otimes I)^T\, H_W\, (A^T \otimes I), \qquad I \in \mathbb{R}^{d\times d}. \tag{22}$$

**Theorem A.11** (Effective Hessian with respect to $A$). *Fix $B$ and consider variations in $A$. For a perturbation $\Delta A$, we have $\Delta W = B\Delta A$. The corresponding effective curvature operator is*

$$H_A(\Delta A) = B^\top H_W(B\Delta A). \tag{23}$$

*Proof.* Substitute $\Delta W = B\Delta A$ into the quadratic form:

$$\frac{1}{2}\langle \Delta W,\ H_W \Delta W \rangle = \frac{1}{2}\langle B\Delta A,\ H_W(B\Delta A)\rangle$$
$$= \frac{1}{2}\langle \Delta A,\ B^\top H_W(B\Delta A)\rangle,$$

which establishes the claim. $\square$

**Corollary A.12** (Operator and matrix forms of $H_A$). *In operator notation,*

$$H_A = B^\top \circ H_W \circ B.$$

*In matrix form,*

$$H_A = (I \otimes B^\top)\, H_W\, (I \otimes B), \qquad I \in \mathbb{R}^{d\times d}. \tag{24}$$

### A.2.2 CONDITION NUMBER BOUNDS

In the main paper we have the following bound of the condition number of $H_A$ in Equation (3):

$$\max\left\{ \frac{\kappa(H_W)}{\kappa(B)^2},\ \frac{\kappa(B)^2}{\kappa(H_W)} \right\} \le \kappa(H_A) \le \kappa(B)^2 \cdot \kappa(H_W). \tag{25}$$

We show this to be true below.

**Lemma A.13** (Upper bound). *For any full-rank $X, Y$,*

$$\kappa(XY) \le \kappa(X)\,\kappa(Y).$$

*Proof.* By definition, $\kappa(XY) = \sigma_1(XY)/\sigma_{\min}(XY)$. Submultiplicativity of the spectral norm gives $\sigma_1(XY) \le \sigma_1(X)\sigma_1(Y)$, and the inequality $\sigma_{\min}(XY) \ge \sigma_{\min}(X)\sigma_{\min}(Y)$ yields the result:

$$\kappa(XY) \le \frac{\sigma_1(X)\sigma_1(Y)}{\sigma_{\min}(X)\sigma_{\min}(Y)} = \kappa(X)\kappa(Y).$$

$\square$

**Lemma A.14** (Lower bound). *For any full-rank $X, Y$,*

$$\kappa(XY) \ge \max\left\{ \frac{\kappa(X)}{\kappa(Y)},\ \frac{\kappa(Y)}{\kappa(X)} \right\}.$$

*Proof.* Using the min–max characterization of singular values:

$$\sigma_1(XY) = \max_{\|u\|=1} \|XYu\| \geq \min_{\|v\|=1} \|Xv\| \cdot \max_{\|u\|=1} \|Yu\| = \sigma_{\min}(X)\,\sigma_1(Y).$$

Similarly,

$$\sigma_{\min}(XY) = \min_{\|u\|=1} \|XYu\| \leq \|X\|_2 \cdot \min_{\|u\|=1} \|Yu\| = \sigma_1(X)\,\sigma_{\min}(Y).$$

Taking the ratio gives

$$\kappa(XY) = \frac{\sigma_1(XY)}{\sigma_{\min}(XY)} \geq \frac{\sigma_{\min}(X)\,\sigma_1(Y)}{\sigma_1(X)\,\sigma_{\min}(Y)} = \frac{\kappa(Y)}{\kappa(X)}.$$

Swapping $X$ and $Y$ yields the other inequality. $\square$

**Theorem A.15** (Bounds for products). *For any full-rank $X, Y$,*

$$\max\left\{\frac{\kappa(X)}{\kappa(Y)},\ \frac{\kappa(Y)}{\kappa(X)}\right\} \leq \kappa(XY) \leq \kappa(X)\,\kappa(Y).$$

Now let $H_W \in \mathbb{R}^{d \times d}$ be symmetric positive definite (SPD), associated with a weight matrix $W = W_0 + AB$ where $B \in \mathbb{R}^{d \times r}$ and $A \in \mathbb{R}^{r \times d}$. Consider the curvature matrices

$$H_A = (I \otimes B)^\top H_W (I \otimes B), \qquad H_B = (A^\top \otimes I) H_W (A \otimes I),$$

where $\otimes$ denotes the Kronecker product.

**Lemma A.16** (Condition number of quadratic form). *If $H_W$ is SPD, then*

$$H_A = Z^\top Z, \quad Z = H_W^{1/2}(I \otimes B).$$

*Hence,*

$$\kappa(H_A) = \kappa(Z)^2.$$

**Theorem A.17** (Condition number bounds for $H_A$). *Assume $H_W$ is SPD and $B$ is full-rank. Then*

$$\max\left\{\frac{\kappa(H_W)}{\kappa(B)^2},\ \frac{\kappa(B)^2}{\kappa(H_W)}\right\} \leq \kappa(H_A) \leq \kappa(H_W)\,\kappa(B)^2.$$

*Proof.* Apply Theorem A.15 with $X = H_W^{1/2}$, $Y = I \otimes B$. Since $\kappa(I \otimes B) = \kappa(B)$, the inequality follows. Squaring both sides yields the result. $\square$

**Corollary A.18** (Condition number bounds for $H_B$). *Assume $H_W$ is SPD and $A$ is full-rank. Then*

$$\max\left\{\frac{\kappa(H_W)}{\kappa(A)^2},\ \frac{\kappa(A)^2}{\kappa(H_W)}\right\} \leq \kappa(H_B) \leq \kappa(H_W)\,\kappa(A)^2.$$

# B  ADDITIONAL RESULTS

## B.1  GRADIENT ANALYSIS LORA AND OP-LORA

Recall that in Eq. (3) of the main paper we show

$$\max\left(\frac{\kappa(H_W)}{\kappa(A)^2},\ \frac{\kappa(A)^2}{\kappa(H_W)}\right) \leq \kappa(H_B) \leq \kappa(A)^2\,\kappa(H_W), \tag{26}$$

where $A$ and $B$ are LoRA matrices, $W$ are the base weights, $\kappa(\cdot)$ denotes the condition number, and $H$ are the corresponding Hessians. Equation (26) implies that LoRA can exhibit worse Hessian conditioning than full finetuning.

A high Hessian condition number indicates very high curvature in some directions relative to others. This matters because the step size must be small enough to avoid instabilities along the highest-curvature direction; the maximal stable learning rate is then limited by that direction and may be too

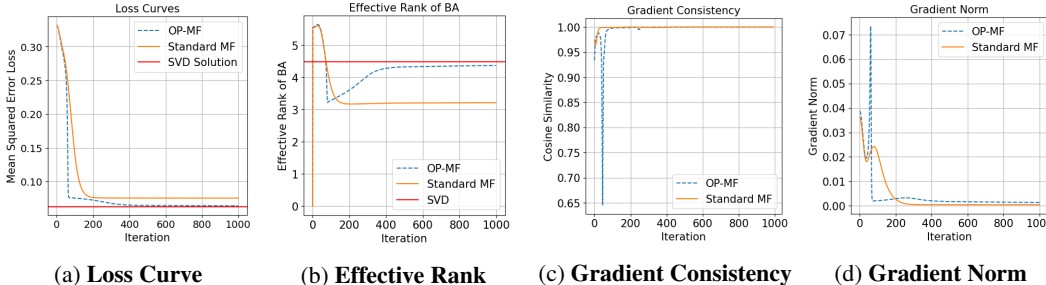

|  (a) **Loss Curve** | (b) **Effective Rank** | (c) **Gradient Consistency** | (d) **Gradient Norm** |

Figure 5: **Matrix Factorization (MF) Gradient Analysis: (a)** Loss Curve shows the reconstruction error for matrix factorization. OP-MF converges better and faster than standard MF. **(b)** Effective Rank of BA reveals changes in the rank of the learned solution. OP-MF learns an effective rank closer to that of the ground-truth SVD solution. **(c)** Gradient Consistency measures the similarity of gradients across iterations. OP-MF is able to make a sudden change in optimization direction, while standard MF cannot. **(d)** Gradient Norm illustrates the scale of gradients. OP-MF is able more quickly adjust optimization step size.

small to make progress in low-curvature directions. This becomes problematic when the highest-curvature direction has already been minimized, but the low-curvature directions still require larger learning rates.

A useful diagnostic is the magnitude of the gradient in the direction of the principal singular vector of the Hessian of the trainable parameters. Let $v$ be that direction (estimated by power iteration) and $g$ the gradient. If $|v^\top g|$ is relatively large, the loss can still be reduced even with a learning rate small enough to remain stable in the largest-curvature direction. Conversely, if $|v^\top g| \approx 0$, a large condition number will prevent further decrease of the loss.

**Setup.** We use the power-iteration method to estimate $v$ and then measure $|v^\top g|$ for a small-scale Rotated-MNIST problem. Pre-training is performed on MNIST and continued training is on Rotated MNIST (as in Fig. 2 of the main paper). We report the terminal values for LoRA and OP-LoRA.

| Method | $|v^\top g|$ |
|---------|---------|
| OP-LoRA | 0.42 |
| LoRA | 0.06 |

Table 5: $|v^\top g|$ at the end of training on Rotated MNIST. Higher is better (indicates remaining descent along the highest-curvature direction).

**Findings.** OP-LoRA exhibits a much larger $|v^\top g|$ in the direction of largest curvature than LoRA (Table 5). Empirically, this suggests OP-LoRA may be *less sensitive* to poor conditioning than LoRA, because it can continue to reduce the loss even when the step size is constrained by the highest-curvature direction. Thus, even if the OP-LoRA MLP were itself poorly conditioned, this sensitivity matters less than for LoRA.

These observations are consistent with the view that OP-LoRA adaptively reshapes the LoRA loss landscape via reparameterization, leading to better and faster optimization.

## B.2 A MATRIX FACTORIZATION CASE STUDY

To verify that the gradient properties of MLP over-parameterization results in observable changes, we design a controlled matrix factorization experiment comparing MLP-generated low-rank matrices $A$ and $B$ with freely learned parameter matrices and measure convergence and gradients.

Matrix factorization decomposes a target matrix $M \in \mathbb{R}^{m \times n}$ into two lower-dimensional matrices, $A \in \mathbb{R}^{r \times n}$ and $B \in \mathbb{R}^{m \times r}$, where $r$ is the latent dimension or rank. This decomposition allows us to

approximate $M$ by $BA$. It can be solved exactly with SVD, or as in our study, one can use gradient descent to minimize the reconstruction error:

$$\|M - BA\|_F^2,$$

where $\|\cdot\|_F$ denotes the Frobenius norm. This resembles LoRA-tuning, where the pre-trained base weights are set to all zeros and the target matrix $M$ is the full finetuning gradient matrix, making it an interesting proxy problem to study.

**Experimental setup and training protocol:** We construct a synthetic target matrix $M \in \mathbb{R}^{100 \times 100}$ with entries initialized uniformly at random from 0 to 1. The resulting matrix has a poor condition number, defined as the ratio between the largest singular value and lowest, making the optimization difficult and therefore a good test for MLP reparameterization. We train for 1000 steps with SGD, with linear warmup for 50 steps and linear learning rate decay.

**OP-MF Model:** The OP-MF model generates matrices $A$ and $B$ through two separate MLPs. We enforce both matrices to be of rank 8. Each MLP receives a learned input vector $z \in \mathbb{R}^{128}$ and processes it through two fully connected layers with 32 hidden units and ReLU activations, outputting the entries for either $A$ or $B$. The second layer is heavily overparameterized; the parameter count is number of hidden units in the MLP by the size of the parameter matrix $A$ or $B$. To align with LoRA's initialization strategy, the MLP for matrix $B$ is initialized to output zeros, setting the model close to a pre-trained state.

**Matrix Factorization(MF) Model:** We train a MF model with freely learnable matrices $A$ and $B$, initialized with random values for $A$ and zeros for $B$. Again, both matrices have rank 8.

**Finding 1: OP-MF rapidly adapts step size and direction.** We examine the gradients, looking for evidence that OP-LoRA adaptively changes step size and direction. Our results reveal that as predicted, OP-MF shows an ability to rapidly adapt step sizes. This can be seen in Figure 5 (d), where the gradient norm experiences a sharp spike, followed by a collapse, corresponding to acceleration and slow down in the loss curve in Figure 5 (a). The sudden phase change in gradient norm also corresponds to a direction change in trajectory, measured by the cosine similarity between gradients at 10-step intervals in in Figure 5 (c). Therefore, MLP reparameterization rapidly changes step size and direction, as suggested by the mathematical analysis in Section 3 of the main paper.

**Finding 2: OP-MF is more effective at reaching the SVD solution than standard MF trained with SGD.** In Figure 5(a), we study the loss curves for both the MF model and the OP-MF model. The plot tracks MSE loss over 1000 iterations. The red line represents the SVD solution as a baseline. Interestingly, OP-MF solutions reach the best-case reconstruction error achieved with SVD, while Standard-MF cannot.

In addition to reconstruction error, another way to track progress towards a solution in matrix factorization is plotting the effective rank of the predicted matrix $BA$ over the course of training. Effective rank $\rho$ is defined as

$$\rho = \exp\left(-\sum_{i=1}^{r} \sigma_i \log \sigma_i\right)$$

where $\sigma_i$ represents the normalized singular values of $BA$, and $r$ is the rank of the matrix. One would expect the effective rank to converge towards the effective rank of the ground-truth SVD solution for successful optimization runs.

In Figure 5 (b), we observe the behavior of effective rank across iterations for both MF and OP-MF. We find that OP-MF can approximate the effective rank of the best-case SVD solution much more closely than standard MF.

**Finding 3: OP-MF composes well with both SGD with Momentum and Adam.** A natural question is if using standard acceleration methods like SGD with Momentum or Adam is enough. In Figure 6 we present experiments by adding Momentum to SGD and replacing it entirely with Adam, an optimizer that combines adaptive learning rates with momentum . Both Adam and SGD with Momentum improve reconstruction error for MF, but neither reach the SVD solution. Moreover, OP-MF composes with even best-case and advanced optimizers to find the SVD solution even faster, indicating their complimentary nature.

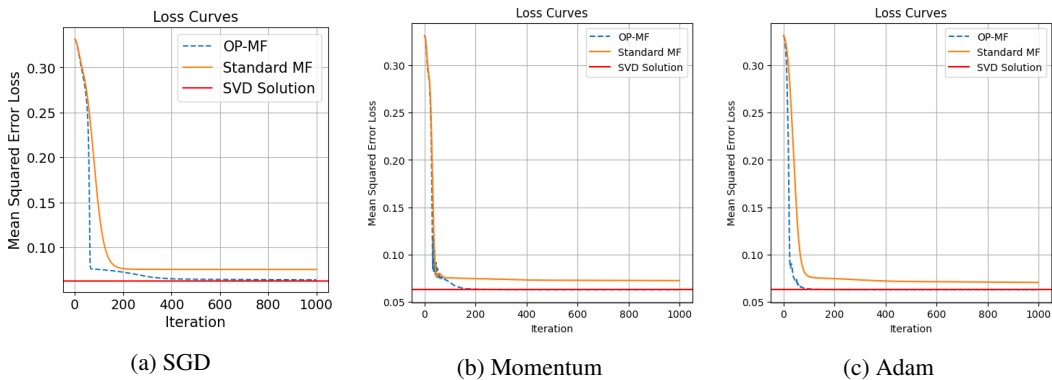

(a) SGD  (b) Momentum  (c) Adam

Figure 6: **Comparing Optimizers.** MLP reparameterization accelerates matrix factorization, even with advanced optimizers.

## B.3 LLaVA Image Classification

Zhang et al. (2024) recently demonstrated that by applying simple finetuning to adapters, large multi-modal models like LLaVA can achieve surprisingly high performance on a variety of classification tasks. We leverage this finding, replacing full finetuning with LoRA and OP-LoRA.

**Datasets:** We use the Stanford Cars (Krause et al., 2013) dataset, which is a fine-grained dataset of about 8000 training examples consisting of 196 classes of cars.

**finetuning and evaluation protocol:** We follow (Zhang et al., 2024), and convert image classification into a captioning task by using the format "⟨ image ⟩ What type of object is in this photo? ⟨ class name ⟩" and training with a language modeling objective. We finetune the visual projector layers of LLaVA1.5-7B for 50 epochs. We set MLP width to 768, since memory resources are not an issue for training only the adapter. At evaluation, we parse the generations and search for the correct class label.

**Results:** We present the results in Table 6. OP-LoRA consistently outperforms LoRA at both rank levels, achieving 83.6% at rank 16 and 87.1% at rank 64, compared to 82.8% and 86.3% with LoRA. This result further reinforces the efficacy of MLP re-parameterization.

| Rank | LoRA | OP-LoRA |
|------|------|---------|
| 16 | 82.8 | **83.6** |
| 64 | 86.3 | **87.1** |

Table 6: LLaVA1.5-7B Image Classification, Top-1 Accuracy on Stanford Cars (Krause et al., 2013).

## B.4 Stability of OP-DoRA

We found that OP-DoRA is more stable than DoRA, as shown with standard deviations across 3 runs. We hypothesize that this is a reflection of decreased learning rate sensitivity.

| Method | Commonsense |
|--------|-------------|
| DoRA ($r = 32$) | $73.7 \pm 6.7$ |
| OP-DoRA ($r = 32$) | $77.5 \pm 1.6$ |

## B.5 VeRA and OP-VeRA

We extend our method to VeRA (Kopiczko et al., 2024), an ultra–low-parameter variant of LoRA, and evaluate on the GLUE(Wang et al., 2018) benchmark using a RoBERTa-base(Liu et al., 2019) backbone following the setup in (Kopiczko et al., 2024). We report results on SST-2, CoLA, and QNLI.

We exclude MNLI due to computational constraints and therefore also excludeMRPC/RTE/STS-B, which are commonly initialized from MNLI to mitigate overfitting (Hu et al., 2022).

| | SST-2 | CoLA | QNLI | **Avg.** |
|---|---|---|---|---|
| VeRA | 94.0 | 59.8 | **91.7** | 81.8 |
| OP-VeRA | **94.3** | **62.1** | 91.5 | **82.6** |

Table 7: GLUE dev results with RoBERTa-base.

Averaged over the three tasks, OP-VeRA improves upon VeRA by **1.2** points, indicating the generality of the proposed optimization.

### B.6 Improving Mix-of-Show with OP-LoRA

Following Zhang & Pilanci (2024b) directly, we apply Mix-of-Show to a small set of 14 training images of Harry Potter. We compare standard ScaledAdamW and OP-LoRA, keeping all training settings from Zhang & Pilanci (2024b). In Figure 7, we generate images from the learned <potter> tokens as a prompt. We see that OP-LoRA better captures the subject of Harry Potter; there are fewer image of two people (not present in the training set) and the shape of the face is more accurate. Moreover, OP-LoRA generates Harry Potter in causal clothing less frequently.

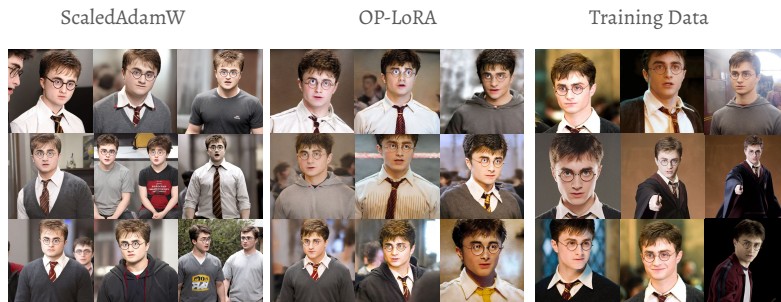

Figure 7: OP-LoRA vs ScaledAdamW(Zhang & Pilanci, 2024b) with Mix-of-Show(Gu et al., 2023).

### B.7 Qualitative Stable Diffusion Results

In the main paper, we show the large quantitative gains OP-LoRA/OP-DoRA give in the image generation task (Table 1). We now present extensive random generations in Figures 8 though 25, with captions from the dataset as input. Several general trends emerge. First, there is a strong color bias towards red, however OP-LoRA and OP-DoRA reduce this dramatically. We attibute this improvement to the over-parameterization easing optimization. Second, over-parameterized LoRA is generates much more diverse and more complex scenes. Overall, qualitative results match the quantitative metrics.

## C Training Details

In this section, we summarize the training settings of our main experiments.

**Code:** We provide an implementation of OP-LoRA and OP-DoRA in the supplementary zip, packaged as a drag-and-drop replacement for existing PEFT libraries. We use this implentation in combination with code from Liu et al. (2024) for CommonSense and VQA benchmarks, von Platen et al. (2022) for Stable Diffusion finetuning, and Zhang et al. (2024) for LLaVA classification.

**Hardware:** Most experiments were done with a single H100 80GB HB3 GPU.

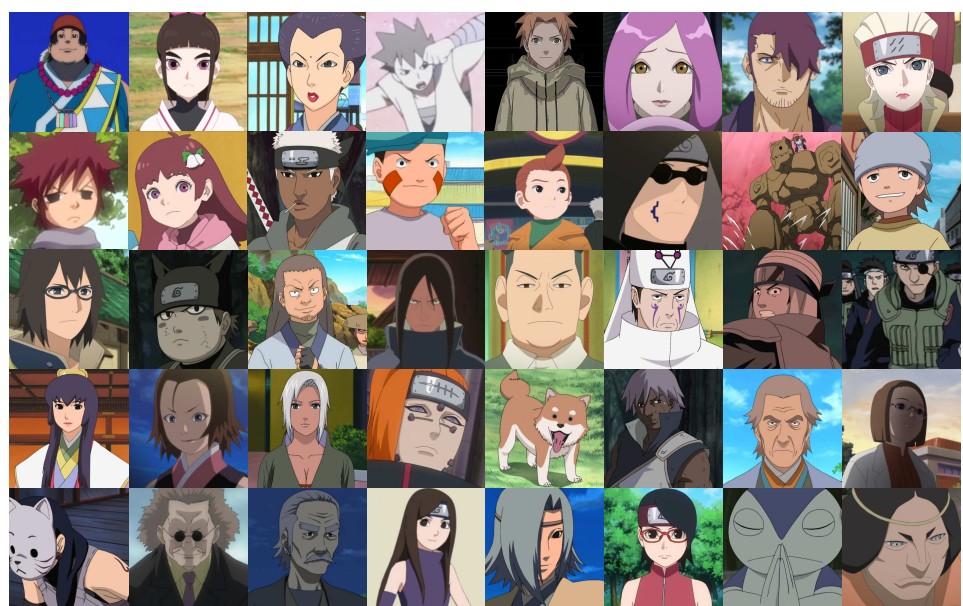

Figure 8: GT Naruto(Cervenka, 2022) images

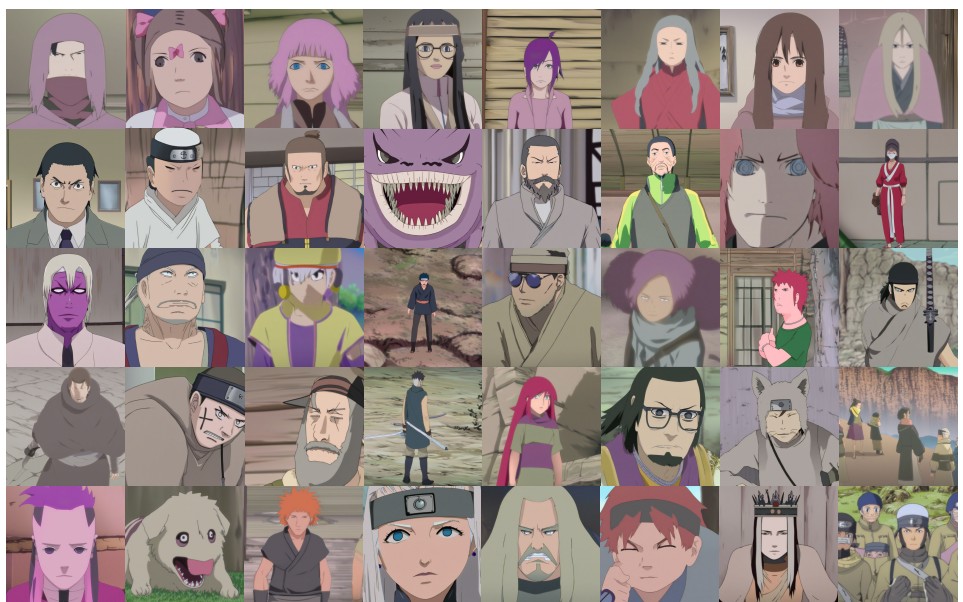

Figure 9: OP-LoRA Naruto(Cervenka, 2022) generated images.

## C.1 Initializing the OP-LoRA MLP

In Section 3, we introduce the MLP used to predict low rank parameters as

$$\begin{pmatrix} A \\ B \end{pmatrix} = W_2(\text{ReLU}(W_1 z + c_1)) + c_2$$

We initialize $W_1$ as Kaiming uniform. We also initalize $W_2$ as Kaiming uniform, except for parameters predicting the upsampling matrix B, which are initialized to zero to make the initialization not change the pre-trained model behavior. All biases are initialized to 0.

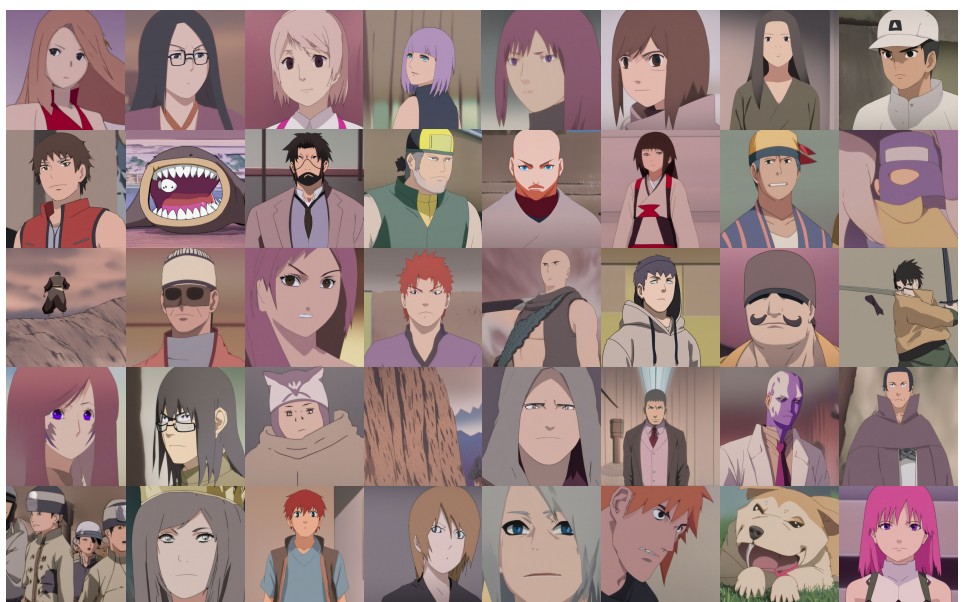

Figure 10: OP-DoRA Naruto(Cervenka, 2022) results

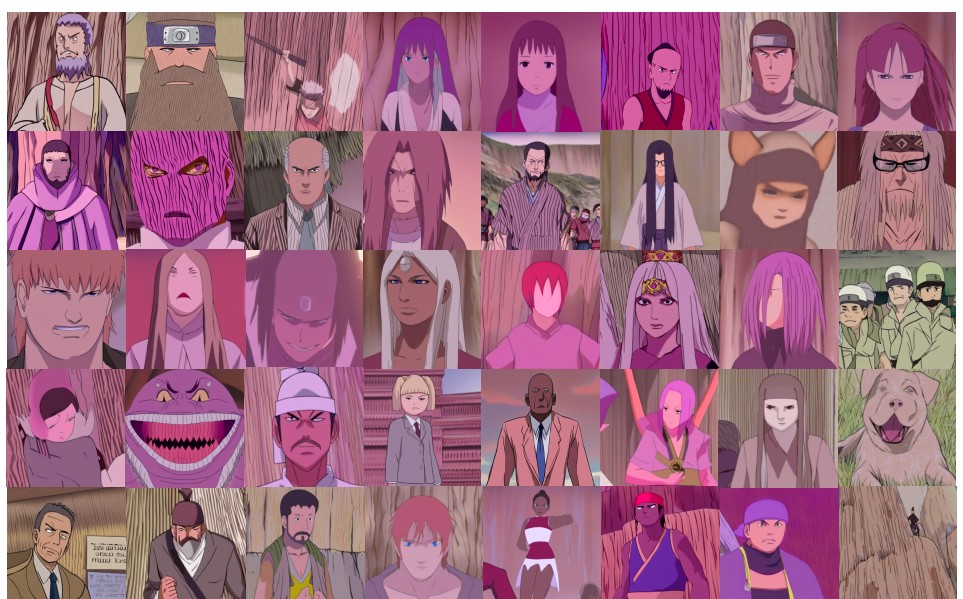

Figure 11: DoRA Naruto(Cervenka, 2022) generated images

## C.2 Training Hyperparameters

In Tables 8 through 11 we show training hyperparameters for experiments. These include batch sizes, learning rates, optimizer settings, and other configurations for each task.

## C.3 LLM Usage

LLM (ChatGPT) was used for polishing during writing, with direct rewrites at the sentence level/equation and suggestions at the paragraph/theorem level. It was also used for verifying notational consistency over the whole paper. All LLM output was carefully reviewed by the authors.

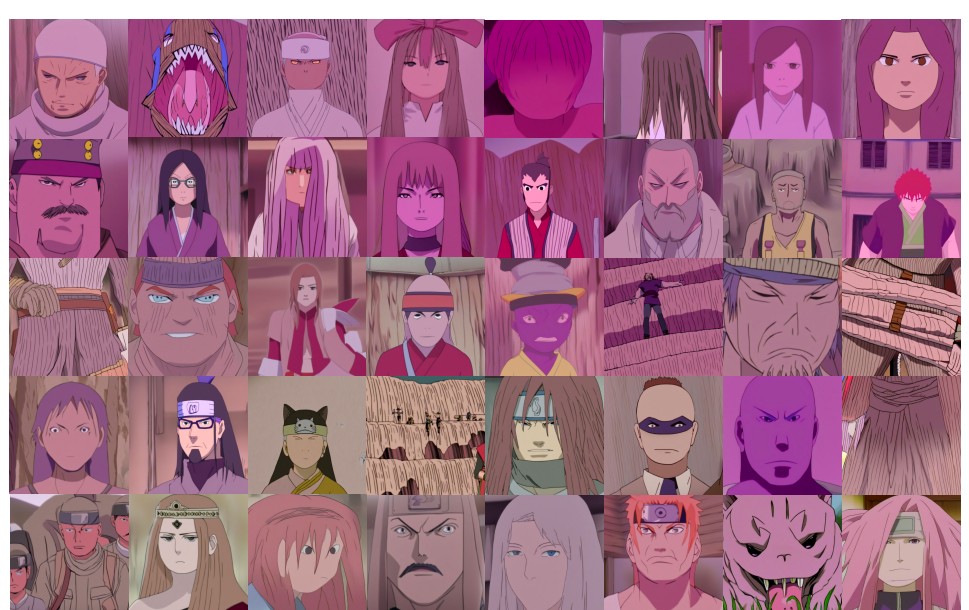

Figure 12: LoRA Naruto(Cervenka, 2022) generated images

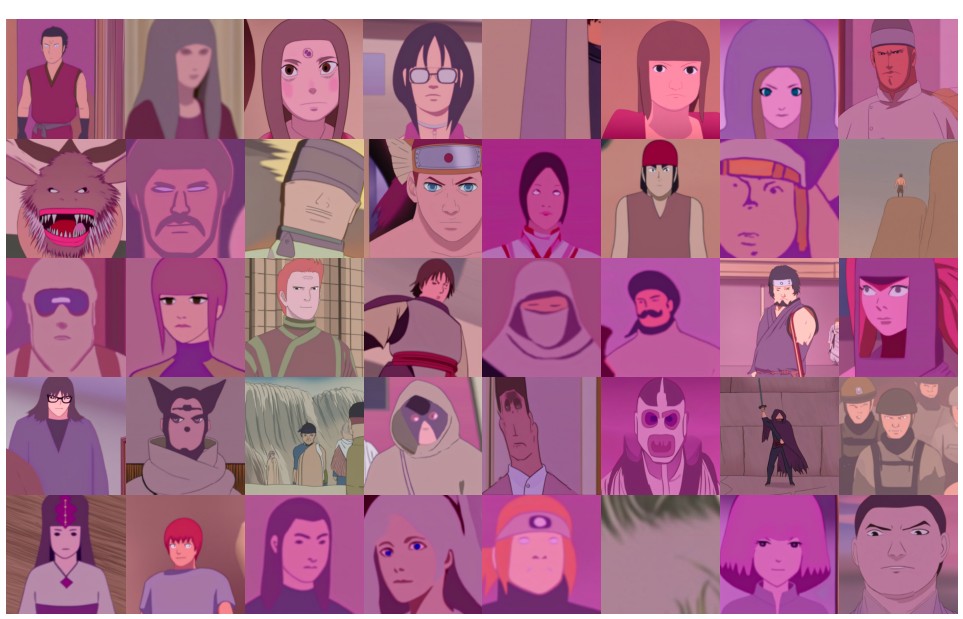

Figure 13: PiSSA Naruto(Cervenka, 2022) generated images

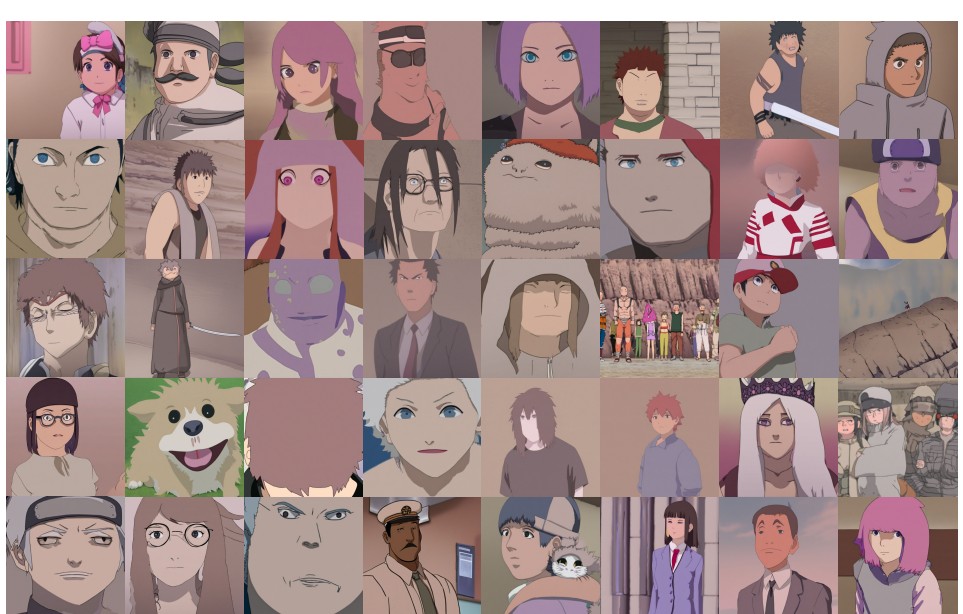

Figure 14: LoRA-GA Naruto(Cervenka, 2022) generated images

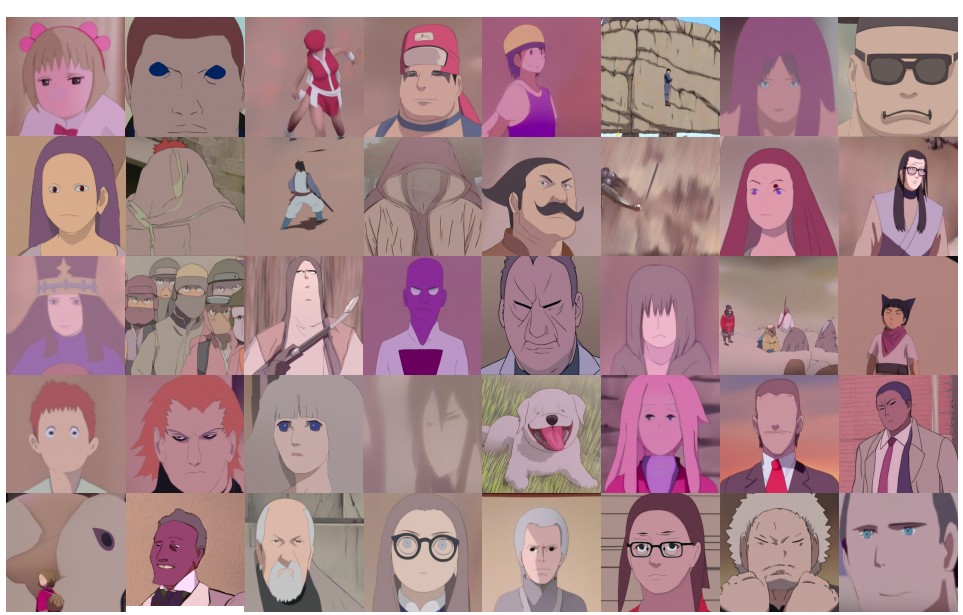

Figure 15: LoRA-PRO Naruto(Cervenka, 2022) generated images

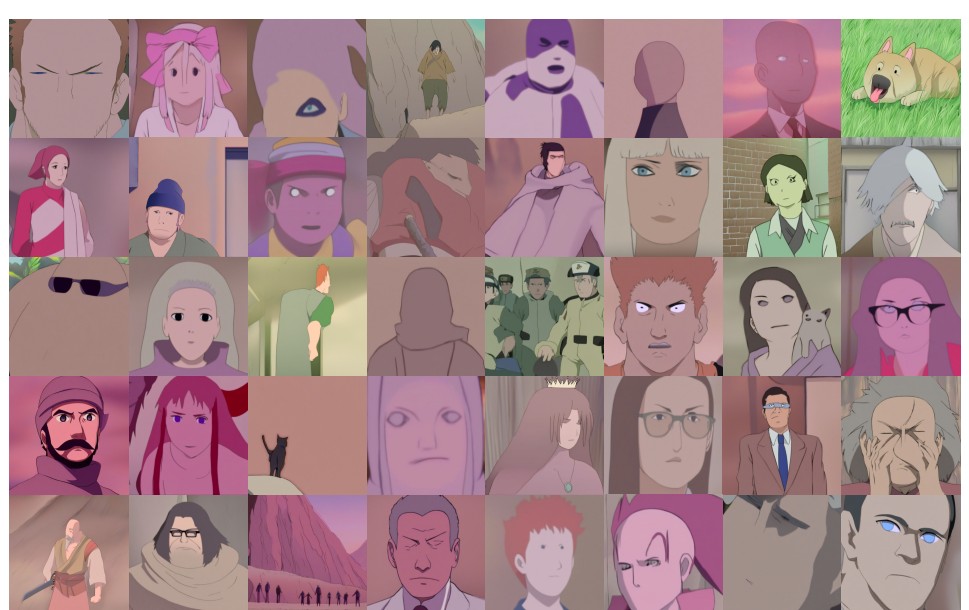

Figure 16: ScaledAdamW Naruto(Cervenka, 2022) generated images

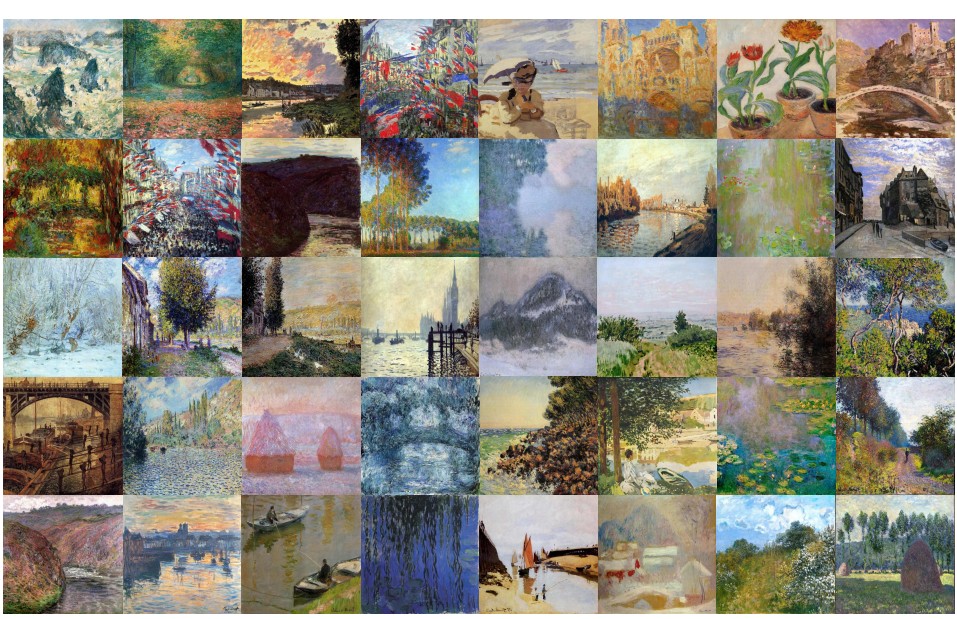

Figure 17: GT Monet WikiArt(Face & Huggan, 2023) Images

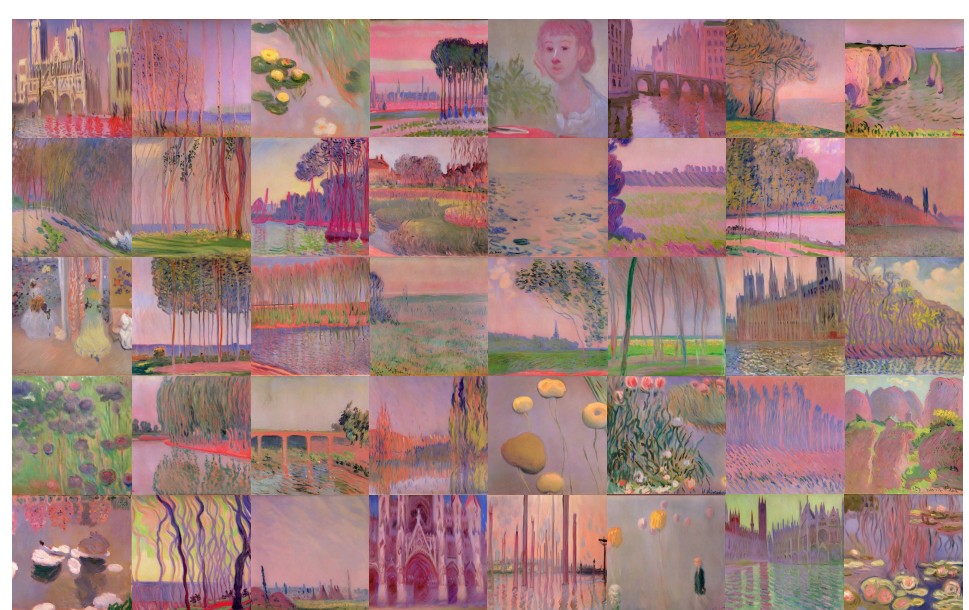

Figure 18: OP-LoRA Monet WikiArt(Face & Huggan, 2023) Generated Images

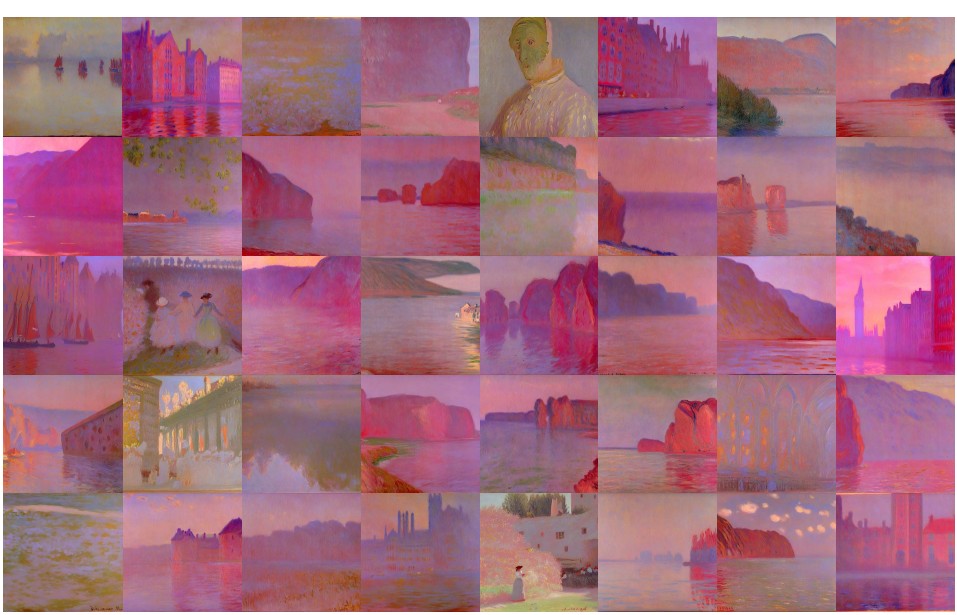

Figure 19: OP-DoRA Monet WikiArt(Face & Huggan, 2023) Generated Images

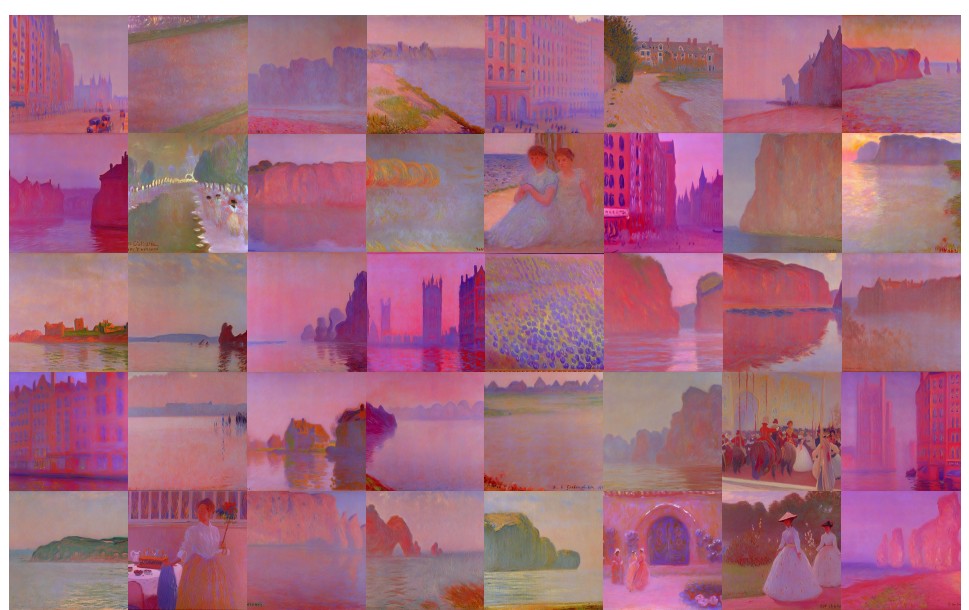

Figure 20: DoRA Monet WikiArt(Face & Huggan, 2023) Generated Images

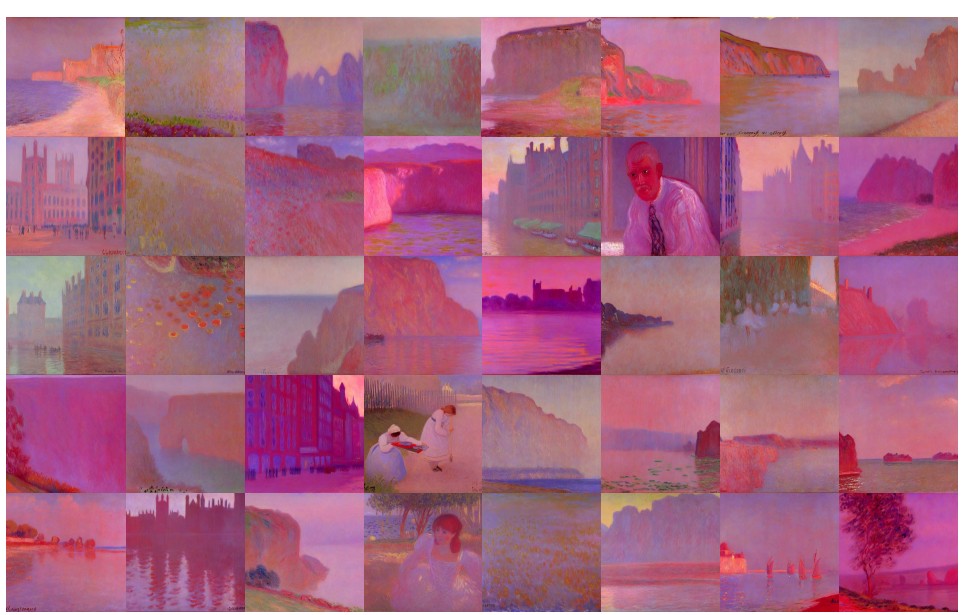

Figure 21: LoRA Monet WikiArt(Face & Huggan, 2023) Generated Images

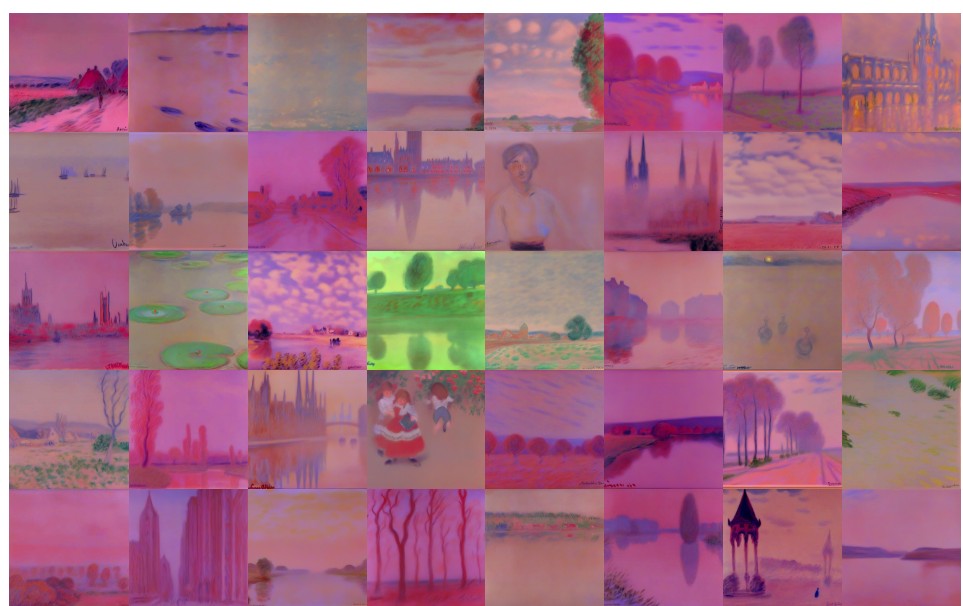

Figure 22: PiSSA Monet WikiArt(Face & Huggan, 2023) Generated Images

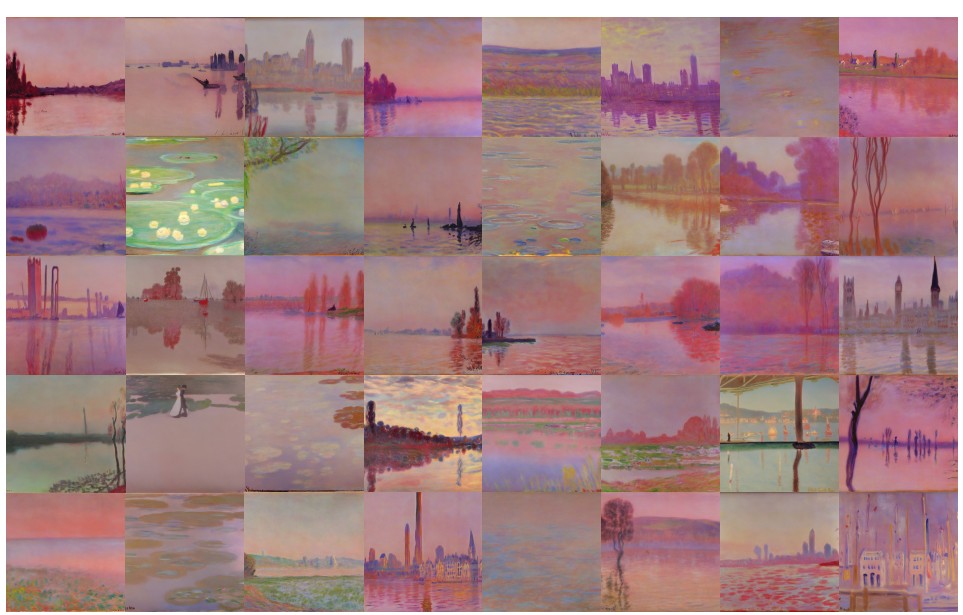

Figure 23: LoRA-GA Monet WikiArt(Face & Huggan, 2023) Generated Images

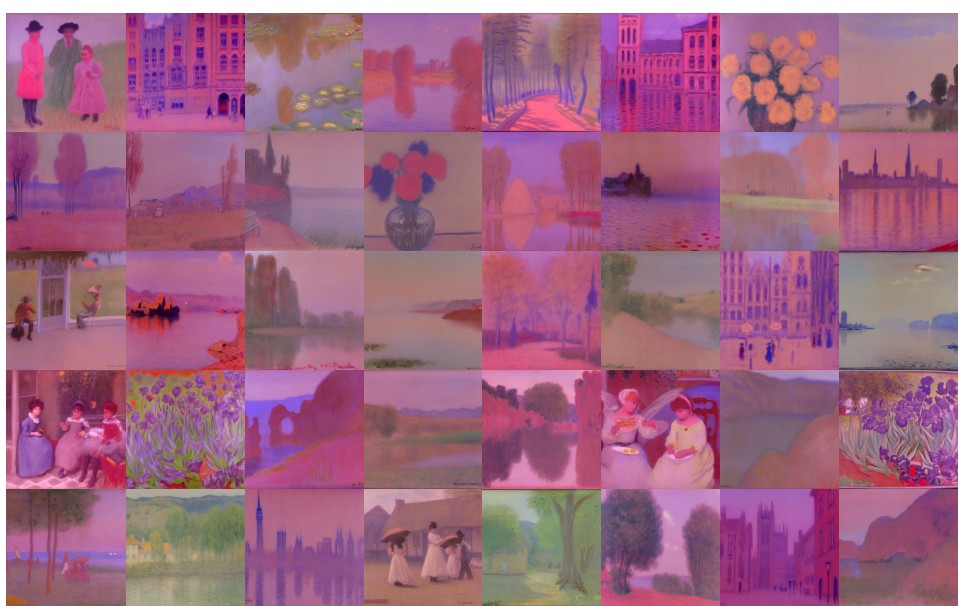

Figure 24: LoRA-Pro Monet WikiArt(Face & Huggan, 2023) Generated Images

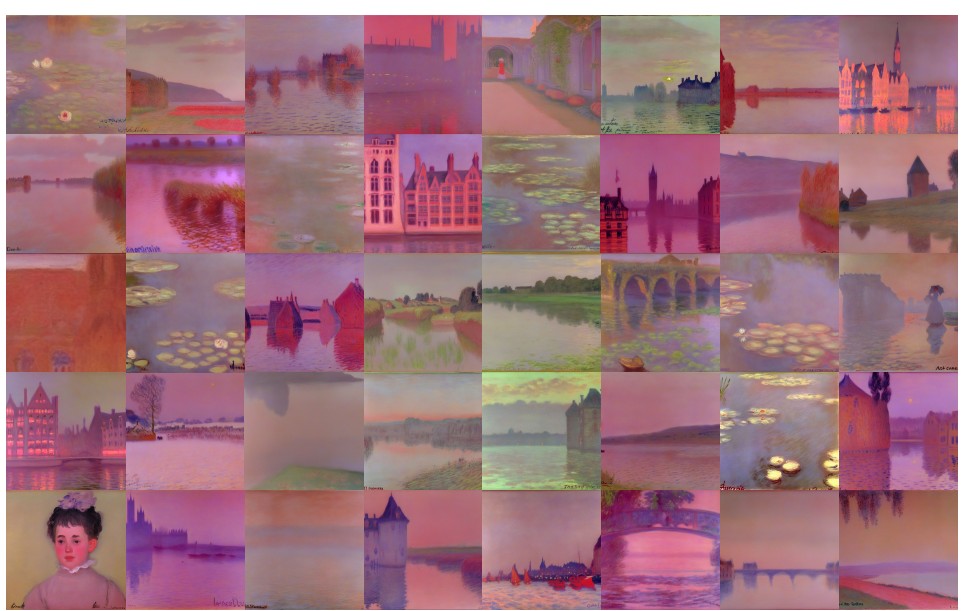

Figure 25: ScaledAdamW Monet WikiArt(Face & Huggan, 2023) Generated Images

| Hyperparameters | All |
| --- | --- |
| Base Model | Stable Diffusion XL 1.0(Podell et al., 2024) |
| Rank $r$ | 4 |
| $\alpha$ | 4 |
| Dropout | 0.0 |
| Optimizer | AdamW |
| LR | 1e-4 |
| LR Scheduler | Constant |
| Batch size | 1 |
| Warmup Steps | 0 |
| Epochs | 2 |
| Where | U-Net Q, K, V, Out |
| MLP-Width(OP-LoRA/OP-DoRA) | 32 |

Table 8: Training Details for Stable Diffusion Finetuning Experiments.

| Hyperparameters | All |
| --- | --- |
| Base Model | VL-Bart(Cho et al., 2021a) |
| Rank $r$ | 128 |
| $\alpha$ | 128 |
| Dropout | 0.0 |
| Optimizer | AdamW |
| LR | 1e-3 |
| LR Scheduler | Linear |
| Batch size | 300 |
| Warmup ratio | 0.1 |
| Epochs | 20 |
| Where | Q,K (Bias Also Trained) |
| MLP-Width(OP-LoRA/OP-DoRA) | 32(OP-LoRA)/4(OP-DoRA) |

Table 9: Training Details for VL-Bart Finetuning Experiments.

| Hyperparameters | All |
| --- | --- |
| Base Model | LLaVA1.5-7B(Liu et al., 2023) |
| Rank $r$ | 64/16 |
| $\alpha$ | 128/32 |
| Dropout | 0.0 |
| Optimizer | AdamW |
| LR | 5e-5 |
| LR Scheduler | Cosine |
| Batch size | 64 |
| Warmup Ratio | 0 .03 |
| Epochs | 50 |
| Where | Multimodal Projector |
| MLP-Width(OP-LoRA/OP-DoRA) | 768 |

Table 10: Training Details for LLaVA Classification Finetuning Experiments.

| Hyperparameters | All |
| --- | :---: |
| Base Model | LLaMA-7B(Touvron et al., 2023) |
| Rank $r$ | 32 |
| $\alpha$ | 64 |
| Dropout | 0.05 |
| Optimizer | AdamW |
| LR | 1e-4 |
| LR Scheduler | Linear |
| Batch size | 16 |
| Warmup ratio | 0.03 |
| Epochs | 3 |
| Where | Q,K, V, Up, Down |
| MLP-Width(OP-LoRA/OP-DoRA) | 32 |

Table 11: Training Details for CommonSense Finetuning Experiments.

