# OpenReview forum: "OP-LoRA: The Blessing of Dimensionality with Overparameterized Low-Rank Adaptation"
_ICLR.cc/2026/Conference — ICLR 2026 Conference Withdrawn Submission_

### Official Review · Reviewer_wGhi · 2025-10-27

**Soundness:** 2
**Presentation:** 3
**Contribution:** 2
**Rating:** 2
**Confidence:** 4

**Summary:**

The paper proposes **OP-LoRA**, a reparameterization of LoRA in which each adapter’s low-rank matrices $A, B$ are predicted at training time by a small MLP. The MLP is discarded after training, so inference storage and compute match standard LoRA. Experiments on multiple tasks and models show that OP-LoRA (and OP-DoRA) outperform their non-overparameterized counterparts and are competitive with gradient-alignment methods such as LoRA-Pro and ScaledAdamW.

**Strengths:**

1. **Novel reparameterization idea.** Using an MLP to generate the LoRA components constitutes a clean reparameterization that is architecture-agnostic and easy to extend.
2. **Consistent empirical gains.** OP-LoRA improves CMMD on SDXL (notably larger gains), and outperforms or matches baselines on other setups.

**Weaknesses:**

1. **Heavy training memory limits practicality.** Table 4 shows OP-LoRA increases GPU memory from 44 GB to 69 GB (+~57%) for LLaMA-7B commonsense finetuning, and extends wall time (3.5 h→4 h). For the paper’s emphasized SDXL customization scenario, where _individual users_ often rely on consumer GPUs, this added VRAM is likely prohibitive, undermining practical utility despite zero inference overhead.
2. **Overfitting narrative is not fully convincing.** The paper asserts OP-LoRA avoids overfitting risks when increasing LoRA rank because representational capacity is unchanged. These are strong claims that would benefit from direct evidence. However, In Table 2, FFT which has more parameters has a higher performance does not suffer from such "overfitting". Table 3 shows rank-related behavior that is not strictly monotone (e.g., LoRA r=466 < r=16; OP-LoRA r=16< r=32).  Recent trends[1-3] also aims to raise the  rank with fixed parameter budgets.
3. **Experimental coverage and fairness.**
- **Missing full-FT baseline.**(Table 1, 3) Full-parameter finetuning is an important reference point and would indirectly test the paper’s assumptions (e.g., the assumption of rank-overfitting problem).
- **Missing related baselines.** Since the paper analyzes LoRA through condition number,  comparisons to similar LoRA improvements would make the case stronger. [4] uses a preconditioning to improve LoRA and [5] develops the same technique from the perspective of condition number.
- **Training-setup transparency.** Baseline details are underspecified. In particular, LoRA baselines typically benefit from larger learning rates; if uniform LR was used without per-method tuning, comparisons may be unfair. Please clarify whether learning-rate sweeps were performed for baselines.

**Overall:** My score lies between 2 and 4. Given the significant training-memory overhead and the concerns above, I lean to 2.

[1] Huang, Q., Ko, T., Zhuang, Z., Tang, L., & Zhang, Y. (2025). HiRA: Parameter-efficient hadamard high-rank adaptation for large language models. In _The Thirteenth International Conference on Learning Representations_.

[2] Sehanobish, A., Dubey, K. A., Choromanski, K. M., Basu Roy Chowdhury, S., Jain, D., Sindhwani, V., & Chaturvedi, S. (2024). Structured unrestricted-rank matrices for parameter efficient finetuning. _Advances in Neural Information Processing Systems_, _37_, 78244-78277.

[3] Jung, Y., Ahn, D., Kim, H., Kim, T., & Park, E. (2025). GraLoRA: Granular Low-Rank Adaptation for Parameter-Efficient Fine-Tuning. _arXiv preprint arXiv:2505.20355_.

[4] Zhang, F., & Pilanci, M. (2024). Riemannian preconditioned lora for fine-tuning foundation models. _arXiv preprint arXiv:2402.02347_.

[5] Zhang, Y., Liu, F., & Chen, Y. (2025). LoRA-One: One-Step Full Gradient Could Suffice for Fine-Tuning Large Language Models, Provably and Efficiently. _arXiv preprint arXiv:2502.01235_.

**Questions:**

1. In Related Work the authors claim: “OLoRA and PiSSA … are prone to overfitting due to removing important components from the frozen base weights.” Could the authors cite specific supporting literature or provide experiments substantiating this?
2. The authors  state LoRA-GA “requires a full-finetuning pass similar to RoSA.” Please specifiy the "full-finetuning pass". RoSA adds a sparse component along with LoRA. LoRA-GA only needs the gradient at $W_0$，where the overhead could be minimized the off-loading.
3. How crucial is the MLP’s nonlinearity? Section 3.2 analyzes a linearized form and moves the MLP extension to the appendix. Could a purely linear generator match performance, or does ReLU materially contribute to the “trainable learning rate / adaptive line search” effects you highlight?
4. Why does LoRA-GA perform worse than vanilla LoRA in Table 3 in your setup? Is this due to implementation choices, hyperparameters, or task-mismatch with SVD-initialized gradients?

---

### Official Review · Reviewer_DHZv · 2025-10-28

**Soundness:** 2
**Presentation:** 3
**Contribution:** 2
**Rating:** 4
**Confidence:** 3

**Summary:**

This paper proposes OP-LoRA, an *over-parameterized reparameterization* of the popular Low-Rank Adaptation (LoRA) method for parameter-efficient fine-tuning.

Instead of directly learning LoRA matrices (A) and (B), OP-LoRA introduces a small MLP hypernetwork that predicts these matrices during training. The MLP is discarded at inference, so inference cost and storage remain identical to standard LoRA.

Experiments across text, vision, and multimodal tasks (LLaMA, VL-BART, Stable Diffusion XL) show consistent improvements over LoRA, and competitive results to heavier methods such as LoRA-Pro and ScaledAdamW, with ~15% training-time overhead.

**Strengths:**

1. **Simple and elegant idea.** The method replaces LoRA’s direct parameters with a small MLP, which is intuitive and requires minimal code changes.
2. **No inference overhead.** The MLP is used only at training time, maintaining the same memory footprint and latency as standard LoRA.
3. **Consistent empirical gains.** Demonstrates improvements of 1–6% (text) and up to 15 CMMD points (image generation) across diverse domains.
4. **Interesting geometric interpretation.** The Hessian-based motivation provides an interpretable theoretical perspective on LoRA’s optimization issues.
5. **Good clarity and presentation.** Figures and writing are clear; the paper’s structure is easy to follow.

**Weaknesses:**

1. **Motivation not fully validated.** The curvature-based explanation (via Hessian conditioning) is plausible but lacks direct evidence such as Hessian spectra or eigenvalue analyses.
2. **High GPU memory usage.** The OP-LoRA requires much higher memory usage than the other LoRA based method (69GB vs 44GB), which strongly limits its deployment on one GPU likes Nvidia RTX 6000. This is my main concern.
3. **Missing ablations** — No experiments isolating the impact of MLP layers size, nonlinearity, the “blessing of dimensionality” claim remains qualitative.

**Questions:**

1. In Table 3, why LoRA (rank=466) has a significant lower performance than other method? What if the performance of LoRA with a slightly higher rank (rank=32, 64, 128)?
2. Have you directly measured or visualized Hessian spectra to validate the curvature argument?
3. Can you include an ablation study separating the effect of overparameterization from the nonlinearity of the MLP?
4. Will the additional parameterization ever lead to overfitting or instability on small datasets?

---

### Official Review · Reviewer_bB1a · 2025-10-30

**Soundness:** 2
**Presentation:** 3
**Contribution:** 2
**Rating:** 4
**Confidence:** 4

**Summary:**

This paper proposes OP-LoRA, a reparameterization strategy that introduces an overparameterized MLP to generate LoRA parameters during training, then discards the MLP at inference time. The method aims to improve optimization stability and learning rate robustness while maintaining zero inference overhead. Experments span multiple tasks and show consistent improvements over standard LoRA and recent variants.

**Strengths:**

- The idea is straightforward and intuitively reasonable. Using overparameterized LoRA only during training mitigates optimization difficulty without increasing inference cost.
- The empirical evaluation covers multiple baselines across vision, vision-language, and language tasks, and the method achieves state-of-the-art performance in most settings.

**Weaknesses:**

- In Table 3, LoRA with rank 466 performs significantly worse than LoRA with rank 16. This contradicts prior observation in the literature, where higher rank generally improves performance. It would be helpful to report results for additional ranks. For example, HiRA [1] shows clear gains at r=512 for commonsense reasoning.
- The method introduces a larger number of training parameters and notably higher GPU memory consumption than LoRA. **This is a core trade-off and brings fairness concerns** when comparing against baselines that operate under much lower compute budget. Including comparisons to full fine-tuning, or initializing LoRA with a high-rank preheating scheme such as HRP [2], would strengthen the claims.

> [1] HiRA: Parameter-Efficient Hadamard High-Rank Adaptation for Large Language Models. ICLR 2025
>
> [2] HRP: High-Rank Preheating for Superior LoRA Initialization. arXiv:2502.07739

**Questions:**

- The paper emphasizes LoRA’s sensitivity to learning rate, yet Figure 2 only evaluates this on Rotated-MNIST, which is much smaller than the main tasks. Could the authors report learning rate sensitivity on the large-scale tasks to provide stronger evidence?
- Were learning rates tuned for LoRA and DoRA? Since OP-LoRA introduces an additional MLP that changes gradient dynamics, using the same learning rate may not ensure fairness. Prior work (including DoRA) suggests that lower learning rates are beneficial. The very weak LoRA/DoRA generated images shown in the appendix raise concerns that suboptimal hyperparameters may be affecting them.
- Why are different ranks chosen across datasets? A unified rank setting or an ablation would help clarify.

---

### Official Review · Reviewer_CqyY · 2025-10-31

**Soundness:** 3
**Presentation:** 3
**Contribution:** 3
**Rating:** 2
**Confidence:** 4

**Summary:**

This paper introduces OP-LoRA which repleaces each LoRA adapter with weights predicted by an extra MLP, an idea from hyper network. This method introduces additional parameters during training while are discard after training, incuring no extra computation during inference. Experiments are conducted to demonstrate the effectiveness of the proposed method, especially on image generation task.

**Strengths:**

1. This paper proposes a novel approach to improving LoRA training, which differs significantly from previous methods.
2. The proposed method is straightforward to implement and generalizes well to other LoRA variants.
3. The paper is well-written and supported by extensive experiments that validate the effectiveness of the approach.

**Weaknesses:**

The paper's contribution appears to be limited. The core idea, leveraging a hypernetwork, is not novel, and its application to LoRA seems straightforward. This limits the overall technical novelty of the work.

Furthermore, the method introduces a significant practical issue: it requires substantially more GPU memory during training. This high memory overhead contradicts a core principle of parameter-efficient fine-tuning (PEFT), which is to be memory-efficient.

Finally, the empirical evaluation reveals a critical limitation. The proposed method shows no significant advantage over standard LoRA in loss when high learning rates are used. This is particularly concerning since LoRA is known to achieve optimal performance with large learning rates, which calls into question the practical utility of the proposed approach.

**Questions:**

see weakness

---

### Note · Authors · 2025-11-14

I have read and agree with the venue's withdrawal policy on behalf of myself and my co-authors.